# CodeRule-RL: Standard-Guided RL with Per-Rule Reward Scheduling for Code LLMs

## Abstract

Large language models for code often pass unit tests yet remain brittle in practice: they may overfit to a test suite, rely on undefined semantics, or fail under small perturbations. Existing RL-based code generation methods optimize rewards from unit test execution, tightly coupling the training signal to a specific test suite. In contrast, we focus on coding standards as the primary source of feedback. We use rules (e.g., MISRA C) as machine-checkable outcomes, converting them into per-rule reward components with a frequency-aware schedule. During reinforcement learning, the model maximizes this rule-based proxy reward. We hypothesize that enforcing well-established coding rules provides a generalizable training signal that improves both adherence to standards and pass@1 (single-attempt functional success). Motivated by these concerns, we present CodeRule–RL, a reinforcement learning approach that integrates coding rules directly as reward signals for code generation. A frequency-aware curriculum prioritizes frequently violated rules and downweights them as compliance improves. The model, optimizer, data, and prompts remain fixed, with training adjusting only reward weights. Unit tests may appear in prompts for specifications, but they are not executed during training. On the public CodeContests+ C subset, CodeRule–RL achieves higher pass@1 while reducing training wall clock time by more than an order of magnitude compared with RL that executes tests during training. Across 1.5B–7B backbones, it consistently improves both coding-standard compliance and functional success, delivering an 87% relative pass@1 gain.

## 1 Introduction

Large language models (LLMs) have significantly advanced code generation, with automated assistants now central to development workflows. However, a gap remains: models often pass unit tests but still exhibit brittle behaviors and may diverge from project-specific constraints, such as coding standards, style guides, and other non-functional requirements.

Functional correctness and rule guidance serve distinct objectives with different levels of tractability. Coding standards aim to improve readability, maintainability, portability, and predictability. However, existing pipelines often provide weak or aggregated supervision. Unit test signals capture functionality but ignore coding practices. At the same time, combining diverse rule outcomes into a single score conflates minor issues with critical constraints. These rules differ in severity, scope, and granularity. They range from expression-level restrictions to translation-unit checks and from naming conventions to constraints on control flow and conversions.

Existing RL-based code generation methods (Wang et al., 2025a) typically optimize rewards based on unit test execution, tightly linking the training signal to specific test suites. In contrast, we prioritize coding standards as the primary source of feedback. We treat rules (e.g., MISRA C) as machine-checkable outcomes and convert them into per-rule reward components, managed by a frequency-aware schedule. This approach directs the model to maximize rule-based proxy rewards during reinforcement learning. We hypothesize that enforcing well-established coding rules offers a generalizable training signal, improving both adherence to standards and pass@1 (single-attempt functional success).

Thus, we introduce *CodeRule-RL*, an RL framework that targets functional correctness as the ultimate evaluation goal, but utilizes coding-standard guidance as an efficient proxy reward signal. Here

*pass@1* is the probability that a single decoded program passes all unit tests in the held-out evaluation suite. Rule-wise verdicts are mapped to reward components so that updates align with individual rules rather than a single aggregated score. A simple frequency aware curriculum schedules rule weights by emphasizing the most frequently violated rule and then expanding to lower-frequency rules as violation rates fall, which focuses updates on the current bottleneck. The curriculum changes only rule weights; data, prompts, and the optimizer remain fixed. Unit test specifications may appear in the prompt to express requirements, but tests are not executed during training.

Our contributions are threefold:

- We formulate coding standard guided RL with per-rule reward shaping and credit assignment that *optimizes a static compliance objective* to implicitly enhance *pass@1*, strictly avoiding the execution of unit tests in the training loop for efficiency.

- We introduce a *frequency aware curriculum* that reweights *per-rule* rewards by empirical violation rates while keeping data, prompts, and the optimizer fixed.

- On a *frozen subset of CodeContests+ (Wang et al., 2025b)*, *CodeRule-RL* improves single-attempt *pass@1* and reduces training time relative to RL that executes tests during training.

## 2 RELATED WORK

### 2.1 LLM-BASED CODE GENERATION

Large language models (LLMs) have rapidly advanced program synthesis, code completion, and general code-quality improvement (Wang et al., 2021; Achiam et al., 2023; Roziere et al., 2023; Shen et al., 2023; Roziere et al., 2023; He & Vechev, 2023; Lozhkov et al., 2024; Xu et al., 2024; Grattafiori et al., 2024; Hui et al., 2024; Agarwal et al., 2025). Most evaluations emphasize *functional correctness* via unit tests and benchmarks (Li et al., 2022; Jain et al., 2024; Wang et al., 2025b; Zhuo et al., 2025), which offer limited guarantees of *coding-standard adherence*. Coding standards encode conventions for predictable, maintainable, and portable software, and major ecosystems provide mature rule sets and linters (e.g., PEP 8/pycodestyle, Checkstyle, ESLint, SwiftLint). Within C, MISRA C defines a disciplined subset that discourages ambiguous constructs (Bagnara et al., 2018; 2021). In our experiments, we use MISRA C:2012 to derive rule-wise guidance signals, while keeping the framework standard-agnostic: any environment with machine-checkable rules can supply the same rule-wise feedback without changing the learning algorithm.

### 2.2 REINFORCEMENT LEARNING FOR CODE LLMs

Prior efforts improve reliability along three lines. *Inference-time guidance* constrains decoding with grammars or type systems and can leverage unit-test feedback (Mündler et al., 2025; Chen et al., 2018; Wang et al., 2025a; Feng et al., 2025; Li et al., 2025), but it does not update the policy, and its effects on adherence are often transient across tasks. *RL from automated feedback (RLAIF)* (Lee et al., 2024; Liu et al., 2023a) updates the model using tool-generated, program-level signals (Dou et al., 2024; Zeng et al., 2025; Dolcetti et al., 2024; Yao et al., 2025). Our approach follows this line but differs in three aspects: (i) objective & setting — we optimize only single-attempt *pass@1* and keep tests out of the training loop; (ii) signal construction — we use bounded *per-rule* checks instead of collapsing heterogeneous findings into a single scalar; (iii) scheduling mechanism — we realize a simple *frequency-aware schedule inside the reward* that adjusts only rule weights over time (no data resampling; optimizer, data, and prompts remain fixed). In ablations, we compare against scalar rewards and per-rule shaping with fixed weights, showing that the in-reward schedule is necessary beyond per-rule shaping alone. We do not construct preference pairs or counterfactual negatives. Unlike RL, which uses unit-test rewards, unit-test specifications appear only in the query, which avoids executing tests during training and improves efficiency. A complementary direction, *neuro-symbolic integration*, couples LLM proposals with formal methods (e.g., SyGuS or verifier-in-the-loop) (Ganguly et al., 2024; Barke, 2024; Chaudhuri et al., 2021; Li et al., 2024; Yan et al., 2025; Jha et al., 2025); in contrast, *CodeRule-RL* integrates per-rule guidance into the policy itself through RL.

Figure 1: **Rule-wise signals and scheduling.** *Left:* Aggregate view of rule-wise signals extracted from a candidate program $C$. *Right:* Reward shaping at the level of individual rules via weights $w_r$ that are scheduled by empirical violation rates. The policy is sampled by prompt; no per-rule resampling or counterfactual rewrites are constructed.

## 3 METHOD

**Problem formulation.** Our objective is to maximize *one-shot functional success* (*pass@1*) while using coding standards only as *auxiliary, structured feedback*. Given a prompt $q$, a policy $\pi_\theta(\cdot \mid q)$ emits a program $C$. Functional correctness is evaluated at test time by $\Gamma(C) \in \{0, 1\}$, where $\Gamma(C) = 1$ iff $C$ compiles and passes the public unit tests. A machine-checkable rule evaluator $\Phi$ maps $C$ to a *rule-wise* signal vector $\mathbf{s}(C) \in \mathbb{N}^m$ for a pluggable rule set $\mathcal{R}$. We define

$$\text{pass@1} = \frac{1}{|\mathcal{X}_{\text{eval}}|} \sum_{x \in \mathcal{X}_{\text{eval}}} \mathbb{1}\!\left(\Gamma(C_x) = 1\right), \tag{1}$$

where each task contributes a single decoded candidate ($k{=}1$). Rule guidance is used purely for reward shaping to improve first-try functional success; it is not an optimization target.

**Symbols and conventions.** $\mathbb{1}[\cdot]$ is the indicator; for $a < b$, $\text{clip}(x, a, b) = \min\{\max\{x, a\}, b\}$. $\mathbb{E}_{C \sim \text{batch}}[\cdot]$ denotes the mini-batch expectation. Vectors are row-stacked by default.

**Motivation.** Unit tests provide functional supervision but are sparse and costly to execute during training. Automatically checkable rule feedback is consistent and fine-grained. We therefore convert $\Phi$'s rule-wise signals into rewards and schedule their influence with a *frequency-aware curriculum* that emphasizes frequently violated rules and gradually rebalances as violation rates decline. Training optimizes only *pass@1*; we do not construct preference pairs or counterfactual negatives.

**Overview.** As shown in Figure 2, a policy $\pi_\theta$ generates a candidate $C$, the evaluator $\Phi$ returns a rule-wise vector $\mathbf{s}(C)$, and this vector is shaped and aggregated into a smooth, bounded reward $R(C, t)$. All curriculum effects are realized *inside the reward* through time- and state-dependent rule weights $w_r(t, \mathbf{s})$ (Sec. 3.4). Data, prompts, and the optimizer remain unchanged. Prompts may include specification text; unit tests are not executed during RL.

### 3.1 DATA PREPARATION

**Sources and splits.** We evaluate on a *frozen* subset of CodeContests+ (Wang et al., 2025b) at the task level, following the official manifest and fixed splits without modification. Prompts are taken directly from the tasks. To prevent cross-split leakage, we apply multi-level decontamination before training and keep the split assignments unchanged. For each task, unit-test suites are curated or normalized and used only at *evaluation* time to define pass@1 and, in ablations, to provide prompt-side specifications; tests are *not* executed during training.

**Rule-wise signals.** A candidate $C$ may trigger multiple rule findings. As shown in Figure 1, we *do not* decompose $C$ into per-rule training samples and we *do not* create compliant rewrites. Instead, we extract a rule-wise signal vector $\mathbf{s}(C)$ and compute a bounded reward by shaping each component at the rule level. A simple frequency-aware schedule updates the per-rule weights to emphasize frequently violated rules and to rebalance as violation rates decline. The online RL sampler always draws inputs by prompt; rule-wise structure is used only in diagnostics and reward computation, not for data resampling.

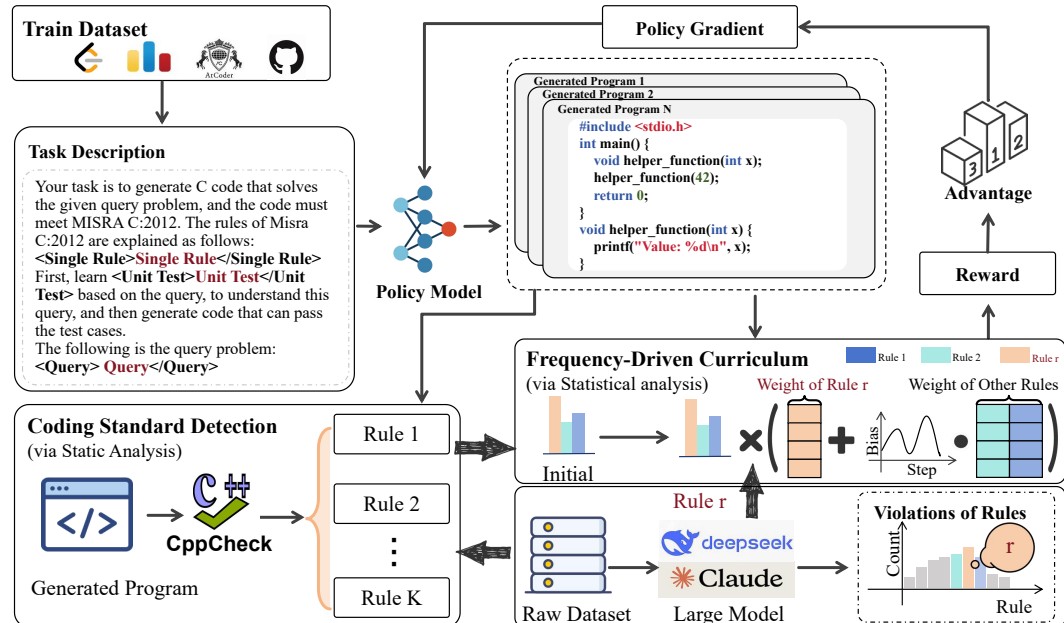

Figure 2: **CodeRule-RL overview.** A policy $\pi_\theta$ emits a candidate program $C$. The evaluator $\Phi$ returns a per-rule vector $\mathbf{s}(C)$. per-rule shaping and scheduled weights $w_r(t, \mathbf{s})$ produce a smooth, bounded reward $R(C, t)$. All curriculum effects are realized inside the reward. Unit tests may appear in prompts but are not executed during RL.

### 3.2 SPEC-TO-REWARD MAPPING

**Rule set abstraction.** We model a coding standard as a finite set $\mathcal{R} = \{r_1, \ldots, r_K\}$. Each rule $r$ is associated with a machine-checkable predicate or counter function $\chi_r : \mathcal{C} \to \mathbb{Z}_{\geq 0}$ that returns a non-adherence score for a program $C$. We treat $\{\chi_r\}$ as black-box oracles and make no assumption about the underlying implementation.

**Per-rule signalization.** Let $a_r \in \{0, 1\}$ indicate whether $r$ is available in the current environment. Define the raw non-adherence vector $\mathbf{u}(C) = \big[u_r(C)\big]_{r \in \mathcal{R}}$ with $u_r(C) = \chi_r(C)$, and an element-wise squashing map $\varphi_r : \mathbb{R}_{\geq 0} \to [0, 1]$ that is monotone and bounded. We form the per-rule signal

$$\mathbf{s}(C) = \big[s_r(C)\big]_{r \in \mathcal{R}}, \qquad s_r(C) = a_r\, \varphi_r\big(u_r(C)\big), \tag{2}$$

which yields a structured, bounded vector suitable for reward shaping and scheduling. This construction is standard-agnostic: replacing the rule set or the checking oracles changes only $\{\chi_r, \varphi_r, a_r\}$ while leaving the learning algorithm unchanged.

### 3.3 REWARD FUNCTION DESIGN

The full algorithmic procedure is presented in Algorithm 1 in the Appendix.

**Rule-guided rewards.** During RL we do not execute unit tests. Rewards are derived solely from *per-rule*, machine checkable verdicts supplied by the rule evaluator $\Phi$. This choice yields (i) deterministic and reproducible feedback independent of runtime behavior, (ii) dense signals even when a candidate fails to compile, (iii) low and predictable latency amenable to batching and caching, and (iv) no exposure of reference test I/O to the objective. For each candidate $C$ we obtain the signal vector $\mathbf{s}(C)$ (Sec. 3.2), shape components into penalties $P_r(\hat{s}_r)$ (Eqs. 3–4), and combine them with curriculum controlled weights $w_r(t, \mathbf{s})$ to produce the bounded reward $R_{\text{rules}}(C, t)$ (Eq. 5).

**Per-rule shaping.** For each rule $r \in \mathcal{R}$, raw scores are capped or smoothed and mapped to a shaped penalty:

$$\hat{s}_r = \psi_r\big(\min\{s_r, N_r\}\big), \qquad \psi_r(x) \in \{x, \sqrt{x}\}, \tag{3}$$

$$P_r(\hat{s}_r) = 1 - \exp(-k_r\,\hat{s}_r), \tag{4}$$

where $N_r$ caps outliers to stabilize gradients, $\psi_r$ selects linear or sublinear smoothing, and $k_r$ controls per-rule sensitivity.

**Aggregation.** With base importances normalized over available rules ($\sum_{r:a_r=1}\bar{w}_r = 1$), the final reward is

$$R_{\text{rules}}(C,t) = \text{clip}\Big(1 - \sum_{r\in\mathcal{R}} w_r(t,\mathbf{s})\,P_r(\hat{s}_r),\ -0.5,\ 1.2\Big). \tag{5}$$

If a valid translation unit cannot be formed or rule checking fails, we assign a fixed penalty $R_{\text{fail}} = -0.5$ (the lower clip bound). The upper bound $1.2$ preserves headroom for KL and entropy terms and keeps the scale numerically stable.

### 3.4 Frequency-Driven Curriculum

**From signals to weights.** The schedule specifies the weights $w_r(t,\mathbf{s})$ used in Eq. 5. Unlike scalar aggregation or data resampling, we prioritize rules by empirical violation frequency and update only the weights inside the reward.

**Ordering and active set.** Let $\bar{s}_r(t)$ be the exponential moving average (EMA) of per-batch rule signals:

$$\bar{s}_r(t) = (1-\lambda)\,\bar{s}_r(t-1) + \lambda\,\mathbb{E}_{C\sim\text{batch}}[s_r(C)], \quad \lambda\in(0,1]. \tag{6}$$

Let $\pi_t$ sort $\{\bar{s}_r(t)\}$ in descending order and define the active set.

$$\mathcal{A}(t) = \text{TopK}_t\big(\{\bar{s}_r(t)\}\big), \qquad K(0) = 1. \tag{7}$$

A rule $r$ is considered *satisfied* when $\bar{s}_r(t) \leq \tau$ for a window of $W$ batches. Typical choices use $\lambda\in[0.01, 0.1]$, a small $K(0)$, and $\tau$ set by the median of early-phase frequencies. We increase $K$ only after all currently active rules are satisfied:

$$K(t+1) = \begin{cases} \min\{K(t)+1,\ K_{\max}\}, & \text{if } \forall r\in\mathcal{A}(t):\ \bar{s}_r(t) \leq \tau \text{ for } W \text{ batches,} \\ K(t), & \text{otherwise.} \end{cases} \tag{8}$$

Ties in $\pi_t$ are broken by a fixed rule index; $K_{\max} = |\{r : a_r = 1\}|$ unless otherwise stated.

**Warmup.** When a rule $r$ enters $\mathcal{A}(t)$ at time $t_r^{\text{on}}$, its weight ramps up over $T_{\text{warm}}$ steps:

$$\alpha_r(t) = \min\Big(1, \frac{t - t_r^{\text{on}}}{T_{\text{warm}}}\Big). \tag{9}$$

**Decay.** Once $r$ is satisfied at $t_r^{\text{sat}}$, its weight decays over $T_{\text{cool}}$ steps:

$$\alpha_r(t) = \max\Big(0,\ 1 - \frac{t - t_r^{\text{sat}}}{T_{\text{cool}}}\Big), \qquad t \geq t_r^{\text{sat}}. \tag{10}$$

An exponential alternative $\alpha_r(t) = \exp\big(-\frac{t - t_r^{\text{sat}}}{T_{\text{cool}}}\big)$ can be used; we keep the linear form in the main text.

**Hysteresis.** To avoid rapid oscillation, a satisfied rule re-enters the active set only if $\bar{s}_r(t) \geq \tau + h$ for a margin $h > 0$.

**Weights and priority masking.** Only active rules receive nonzero weights; newly activated rules are ramped; lower-ranked rules are masked until higher-ranked ones clear within the same sample:

$$w_r(t,\mathbf{s}) = \bar{w}_r\,a_r\,\alpha_r(t)\,\mathbb{1}[r\in\mathcal{A}(t)]\,p_r(t,\mathbf{s}), \tag{11}$$

$$p_r(t,\mathbf{s}) = \mathbb{1}\Bigg[\sum_{u\in\mathcal{H}(t,r)} s_u \leq \epsilon_p\Bigg], \qquad \mathcal{H}(t,r) = \{u\in\mathcal{A}(t) : \text{rank}_{\pi_t}(u) < \text{rank}_{\pi_t}(r)\}, \tag{12}$$

with threshold $\epsilon_p \geq 1$ (we use $\epsilon_p = 1$). Here $\bar{w}_r$ are base importances normalized over available rules, $a_r$ is the availability indicator (Sec. 3.2), and $p_r$ enforces within-sample precedence to reduce credit leakage.

### 3.5 POLICY OPTIMIZATION

We adopt Group Relative Policy Optimization (GRPO) (Shao et al., 2024). For each input, we sample $N=8$ candidates, compute centered advantages from $R_{\text{rules}}(C, t)$, and update the policy with a PPO-style clipped objective (clipping $\epsilon=0.2$) plus a KL penalty $\beta \text{KL}(\pi_\theta\|\pi_{\text{ref}})$ to a frozen reference ($\beta=0.05$). We add a small entropy bonus (0.001) to sustain exploration. Rewards derive solely from per-rule checks and are computed in a *consistent*, batched, and cacheable manner; unit tests are not executed during training.

## 4 EXPERIMENT

**Training dataset.** Using the pipeline in Sec. 3.1 and Appendix Figure 6, we curate a compact, single-turn instruction corpus for RL. We aggregate public prompts and C-focused tasks from general sources and apply light, model-agnostic filtering for compilability and basic formatting. Rule-related feedback is *not* baked into the dataset: we do not construct compliant–noncompliant pairs or counterfactual rewrites, and we do not store analyzer diagnostics as labels. Instead, per-rule signals are computed *online* during RL by the rule evaluator (Sec. 3.2) and used solely for reward shaping. Each training item stores the prompt and minimal metadata; unit tests are reserved for evaluation and, in ablations, for prompt-side specification text only.

**Baseline models.** We evaluate *CodeRule-RL* on two open-source code-LLM families. Within the Qwen line we use *Qwen2.5-Coder-Instruct* (Hui et al., 2024) at 1.5B, 3B, and 7B parameters, covering compact to mid-sized deployments where latency and memory trade against accuracy. Within the DeepSeek line we use *DeepSeek-Coder-Instruct* (Guo et al., 2024) at 1.3B and 6.7B, providing an additional architecture, tokenizer, and pretraining mix. This setup enables a controlled study of whether *CodeRule-RL* yields consistent gains beyond a single backbone. We also include three post-training code LLM baselines: AZR-Coder-3B (reinforced self-play without external supervised data) (Zhao et al., 2025), NextCoder-7B (built on Qwen2.5-Coder and fine-tuned with selective knowledge transfer on synthetic and real edit data) (Aggarwal et al., 2025), and Seed-Coder-8B (model-centric data curation with SFT and preference optimization) (ByteDance Seed et al., 2025). We evaluate the official checkpoints in our unified harness (greedy $T=0$, $k=1$; unified prompts; identical context limits and stop sequences; same GCC/Clang toolchain); details are in Appendix D.

**Implementation and hyperparameters.** We regularize with a KL penalty of 0.05 to a frozen reference and add a small entropy bonus of 0.001. Prompts are capped at 512 tokens and responses at 1024. Training uses a global batch of 1024 (micro-batch 32 across 8 GPUs), a learning rate of $1 \times 10^{-6}$, and 80 steps. Unit tests are not executed during training. We set *per-rule* base importances $\{\bar{w}_r\}$, sensitivities $\{k_r\}$, and caps $\{N_r\}$. Newly activated rules *ramp up* over $T_{\text{warm}}$ steps (Eq. equation 9); after satisfaction they *decay* over $T_{\text{cool}}$ (Eq. equation 10). The frequency-driven curriculum maintains an EMA with rate $\lambda$ (Eq. equation 6), activates the $\text{TopK}_t$ rules at each step (Eq. equation 7), and advances when the EMA $\bar{s}_r(t)$ of all active rules falls below threshold $\tau$ for $W$ batches. Unless otherwise stated, $T_{\text{warm}}=30$, $T_{\text{cool}}=30$, $\lambda=0.3$, $K(0)=1$, and $\tau=0.10$; details are in Appendix D,and I- H

**Task and rationale for the standard.** We study *C code generation*: models read problem statements and emit single translation unit C11 programs that use standard I/O and compile with a fixed toolchain. Given this task focus on C, we adopt *MISRA C:2012* as the reference coding standard because it is widely used in safety-critical software and has mature static-analysis support, enabling scalable and reproducible auditing. We report only *Mandatory* and *Required* findings; *Advisory* items are logged but do not affect compliance summaries (Appendix Table 11 lists the rule set and checker configuration). Required Rule 21.6 is excluded because our single translation unit tasks rely on `stdio.h` for I/O; accordingly, it is omitted from checking and reporting. Compliance is measured offline as a secondary diagnostic and does not influence training rewards.

**Benchmark construction and split hygiene.** We evaluate on a *frozen subset* of CodeContests+ (2025b) using the official manifest and fixed splits *without modification*. The subset primarily contains easy to medium Codeforces problems expressed as single translation unit C with standard I/O and deterministic reference checkers. To control leakage, we apply the official CodeContests+ decontamination pipeline and reuse the published blocklists and fingerprints to screen both training corpora and generated outputs.

Table 1: Comparison of base models and *CodeRule-RL* variants on compliance@1, pass@1, and joint@1. *CodeRule-RL* consistently improves all three metrics, turning near-zero compliance into strong compliance, increasing joint@1 across 1.5B to 7B models, and improving pass@1.

| Model | join@1 (%) | compliance@1 (%) | pass@1 (%) |
|---|---|---|---|
| AZR-Coder-3b (Zhao et al., 2025) | 0.00 | 0.85 | 15.74 |
| NextCoder-7B (Aggarwal et al., 2025) | 0.00 | 1.70 | 36.60 |
| Seed-Coder-8B (2025) | 0.00 | 0.85 | 37.45 |
| Deepseek-Coder-1.3B (2024) | 0.00 | 0.85 | 2.13 |
| Deepseek-Coder-1.3B w / *CodeRule-RL* | 2.64 (**+2.64**) | 2.55 (**+1.70**) | 6.00 (**+3.87**) |
| Deepseek-Coder-6.7B (2024) | 0.00 | 0.00 | 18.72 |
| Deepseek-Coder-6.7B w / *CodeRule-RL* | 11.91 (**+11.91**) | 36.17 (**+36.17**) | 28.09 (**+9.37**) |
| Qwen2.5-Coder-1.5B (2024) | 0.00 | 0.43 | 2.55 |
| Qwen2.5-Coder-1.5B w / *CodeRule-RL* | 4.26 (**+4.26**) | 41.28 (**+40.85**) | 11.49 (**+8.94**) |
| Qwen2.5-Coder-3B (2024) | 0.00 | 0.43 | 20.43 |
| Qwen2.5-Coder-3B w / *CodeRule-RL* | 9.80 (**+9.80**) | 17.45 (**+17.02**) | 22.13 (**+1.70**) |
| Qwen2.5-Coder-7B (2024) | 0.00 | 0.00 | 21.13 |
| Qwen2.5-Coder-7B w / *CodeRule-RL* | **24.26 (+24.26)** | **55.74 (+55.74)** | **39.57 (+18.44)** |

**Metrics.** Our primary metric is *pass@1, compliance@1, and joint@1* . For a fixed evaluation set $\mathcal{X}_{\text{eval}}$ and a single decoded candidate per task ($k{=}1$), we compute

$$\text{pass@1} = \frac{1}{|\mathcal{X}_{\text{eval}}|} \sum_{x \in \mathcal{X}_{\text{eval}}} \mathbb{1}\big(\Gamma(y_x) = 1\big), \tag{13}$$

$$\text{compliance@1} = \frac{1}{|\mathcal{X}_{\text{eval}}|} \sum_{x \in \mathcal{X}_{\text{eval}}} \mathbb{1}\big[\mathcal{C}_{\text{std}}(y_x) = 1\big], \qquad y_x = \text{Decode}_{k=1}(\pi_\theta, x), \tag{14}$$

$$\text{joint@1} = \frac{1}{|\mathcal{X}_{\text{eval}}|} \sum_{x \in \mathcal{X}_{\text{eval}}} \big[\mathbb{1}\big(\Gamma(y_x) = 1 \wedge \mathcal{C}_{\text{std}}(y_x) = 1\big], \tag{15}$$

where $\Gamma(y_x){=}1$ iff the program compiles and passes all tests. Decoding is treated as a factor and evaluated *separately* under (i) greedy ($T{=}0$) and (ii) nucleus ($p{=}0.9$, $T{=}0.6$), both with $k{=}1$; unless otherwise stated, main tables report the greedy setting and nucleus results appear in the appendix. We do not average across settings nor select the better of the two. For each setting we run five independent trials with different random seeds and report mean±SD across trials. Compilation uses GCC13 and Clang17 with comparable C11 flags (`-std=c11 -O2` with strict warnings); any disagreement counts as failure. *pass@1* is the primary outcome in the main text. We ensure that the test prompts used in evaluation do not include any test case information, such as example inputs or expected outputs, to prevent any potential leakage of test-specific details into the decoding process. Specifically, the prompts contain only the problem statement and I/O contract, ensuring fairness in the evaluation across all methods.

### 4.1 MAIN RESULTS

**Effectiveness of *CodeRule-RL*.** Across all backbones (Table 1), *CodeRule-RL* yields consistent absolute gains in **pass@1**: Qwen2.5−Coder−7B 21.13 → 39.57 (+18.44; +87.3%), Qwen2.5−Coder−3B 20.43 → 22.13 (+1.70; +8.3%), Qwen2.5−Coder−1.5B 2.55 → 11.49 (+8.94; +350.6%), DeepSeek−Coder−6.7B 18.72 → 28.09 (+9.37; +50.1%), DeepSeek−Coder−1.3B 2.13 → 6.00 (+3.87; +181.7%). Gains are largest in absolute terms on mid-sized models (7B/6.7B) and largest in relative terms on smaller models (1.5B/1.3B), suggesting that *per-rule reward shaping with a frequency-aware schedule* is especially helpful under limited capacity yet remains effective for larger backbones. We observe no regressions on any backbone. Improvements hold from 1.3B/1.5B to 7B parameters and across two code-model families (Qwen, DeepSeek), indicating that *per-rule rewards with frequency-aware scheduling* generalize beyond a

single backbone and decoding setting. The policy reduces brittle patterns that commonly lead to compile or run failures under stricter warnings or alternative toolchains, which benefits pass@1. As shown in Table 1, in stark contrast to the incremental gains observed in pass@1, *CodeRule-RL* delivers breakthrough improvements in both joint@1 and compliance@1. Various instruct models, in their init state, universally fail to meet established coding standards: they exhibit 0.00% joint@1 and negligible compliance@1 ($\leq 0.85\%$), reflecting a complete lack of adherence to critical coding rules. The integration of *CodeRule-RL* effectively addresses this critical gap, unlocking the models' capability to follow strict coding standards that were previously overlooked. Specifically, Qwen2.5-Coder-7B achieves 55.74% compliance@1 and boosts joint@1 to 24.26% with *CodeRule-RL*. Similarly, DeepSeek-Coder-6.7B compliance@1 surges from 0.00% to 36.17%, while smaller models (1.3B–3B parameter scales) also realize substantial leaps from near-zero baselines: joint@1 climbs to 2.64%–9.80%, and compliance@1 reaches 2.55%–41.28%. These results confirm that *CodeRule-RL* precisely aligns generated content with complex coding constraints without compromising functional correctness. On the contrary, pass@1 concurrently improves by 1.70–18.44 percentage points across all backbones, demonstrating the method's ability to achieve a dual optimization of compliance and correctness, a critical advantage for practical coding scenarios requiring both adherence to standards and reliable functionality. The detailed mean±SD results are presented in Table 13.

**Comparison with post-trained SOTA code LLMs.** We compare against size-matched baselines using a unified evaluation protocol that employs greedy decoding ($T=0$, $k=1$), unified prompts, identical context limits, unified stop sequences, five fixed seeds, and the same GCC/Clang toolchain; unless noted otherwise, all re-evaluated baselines follow this protocol, with details in Appendix D. At the ∼7B tier, *CodeRule-RL* on Qwen2.5-Coder-7B attains **39.57**, surpassing NextCoder-7B (Aggarwal et al., 2025) at 36.60 (+2.97%) and Seed-Coder-8B at 37.45 (+2.12%). At the ∼3B tier, *CodeRule-RL* on Qwen2.5-Coder-3B reaches **22.13**, exceeding AZR-Coder-3B (Zhao et al., 2025) at 15.74 (+6.39). These size-matched results show that *rule-wise reward shaping during RL* translates into higher one-shot functional success than strong post-trained baselines of similar size. Our claims of superiority are restricted to models re-evaluated in our unified harness. However, the SOTA baselines completely fail to satisfy the imposed coding standards(compliance@1). AZR-Coder-3B, NextCoder-7B, and Seed-Coder-8B all yield **0.00% joint@1**, constrained by negligible compliance rates (ranging from 0.85% to 1.70%). In stark contrast, *CodeRule-RL* unlocks significant joint success, propelling Qwen2.5-Coder-7B to **24.26% joint@1** (supported by 55.74% compliance) and the 3B model to **9.80% joint@1**. These results demonstrate that while current SOTA post-training methods effectively optimize for correctness, they neglect the structural constraints required for standard-compliant code generation, a critical capability that *CodeRule-RL* effectively restores.

**Qualitative example.** Figure 3 contrasts a typical base decode with the *CodeRule-RL* output on a CodeContests+ task. The base decode compiles and passes tests under the default harness but uses the nonprototype form of `main` and ignores library return values. After training, the policy produces `int main(void)` and acknowledges I/O returns, while preserving problem logic and I/O behavior. In aggregate, such targeted, *per-rule* adjustments correlate with higher **pass@1** (Table 1); we report compliance only as a secondary observation.

## 4.2 ABLATION STUDY

**Comparison with unit test-based RL (pass@1).** To isolate the effect of the reward signal, we evaluate *CodeRule-RL* and CURE (Wang et al., 2025a) under a prompt-parity protocol on `Qwen2.5-Coder-3B` using the same split, decoding settings, and context limits. *CodeRule-RL* attains the highest **pass@1** (22.13% vs. 19.15% for CURE; base 20.43%; Table 2). In the *No-Test* setting, where all test-specific text is removed from the prompts, *CodeRule-RL* still reaches 20.85%, showing that gains do not depend on prompt-side exposure to tests. Additionally, the experiment incorporating unit tests resulted in a pass@1 of 18.72%, further demonstrating that the improvement in coding standard correctness is independent of unit testing. Beyond accuracy, *CodeRule-RL* trains without executing unit tests: rewards are dense, deterministic, and per rule, which removes harness execution overhead and flakiness, improves credit assignment, and simplifies scaling compared with execution-driven RL. These properties make the source of improvement explicit and confer practi-

**Qwen2.5-Coder-7B-Instruct**

```c
#include <stdio.h>
#include <string.h>
int main() {
    char input[1001];
    scanf("%s", input);
    int len = strlen(input);
    int dot_index = -1;
    for (int i = 0; i < len; i++) {
        if (input[i] == '.') {
            dot_index = i;
            break;
        }
    }
    int last_digit = input[dot_index - 1] - '0';
    if (last_digit != 9) {
        int rounded_up = last_digit + 1;
        for (int i = dot_index - 2; i >= 0; i--) {
            int current_digit = input[i] - '0';
            if (current_digit + 1 >= 10) {
                input[i] = '0';
            } else {
                input[i] = current_digit + 1 + '0';
                break;
            }
        }
        input[dot_index - 1] = rounded_up + '0';
        for (int i = dot_index; i < len; i++) {
            input[i] = '0';
        }
        printf("%s", input);
    } else {
        printf("GOTO Vasilisa.");
    }
    return 0;
}
```

**Ours**

```c
#include <stdio.h>
#include <string.h>
int main(void) {
    char number[1001];
    (void)scanf("%s", number);
    int len = strlen(number);
    int dot_index = -1;
    for (int i = 0; i < len; i++) {
        if (number[i] == '.') {
            dot_index = i;
            break;
        }
    }
    int last_dot_index = dot_index - 1;
    int next_dot_index = dot_index + 1;
    int last_digit = number[last_dot_index] - '0';
    int next_digit = number[next_dot_index] - '0';
    if (last_digit != 9 && next_digit < 5) {
        (void)printf("%.*s", dot_index, number);
    } else if (last_digit != 9) {
        number[dot_index - 1] = (last_digit + 1) + '0';
        (void)printf("%.*s", dot_index, number);
    } else {
        (void)printf("GOTO Vasilisa.");
    }
    return 0;
}
```

**[misra-c2012-8.2]**
Function types shall be in prototype form with named parameters.

**[misra-c2012-17.7]**
The value returned by a function having non-void return type shall be used.

Figure 3: An off-the-shelf output versus the *CodeRule-RL* output on a CodeContests+ task.

cal advantages in computational cost and training stability. Details of the experimental design are provided in Appendix J.

**Training efficiency and design factors.** *CodeRule-RL* trains in **7.08** hours, compared to **21.56** hours for CURE ($\sim 3\times$ faster) and 86.67 hours for *CodeRule-RL* with unit tests ($\sim 12\times$ faster). Furthermore, it achieves a significantly lower average reward latency per program (**0.69** s), reducing latency by $\sim 9\times$ compared to CURE (6.36 s) and by $\sim 45\times$ compared to the unit-test-enhanced version (31.21 s).

Rewards depend solely on *per-rule* checks from the evaluator introduced in Sec. 3.2 and are computed consistently in batched, cacheable form without executing unit tests during training. A simple frequency-aware schedule inside the reward adjusts only rule weights over time; data, prompts, and the optimizer remain fixed.

Table 2: Effect of executing unit tests during training on `Qwen2.5-Coder-3B`. The primary metric is **pass@1**. We compare the base model, CURE (execution-based RL), and *CodeRule-RL*. We also report wall-clock training time (hours) and average reward computation latency (s/sample).

| Model | pass@1 (%) | Training time (h) | Latency (s/sample) |
|---|---|---|---|
| Qwen2.5-Coder-3B | 20.43 | – | – |
| CURE (Wang et al., 2025a) | 19.15 | 21.56 | 6.36 |
| *CodeRule-RL* w/o unit-test prompt | 20.85 | – | – |
| *CodeRule-RL* w/ unit-test | 18.72 | 86.87 | 31.21 |
| *CodeRule-RL* | **22.13** | **7.08** | **0.69** |

**Curriculum vs. All Rules.** As shown in Figure 4, a frequency-aware schedule over *per-rule* rewards raises **pass@1** more quickly and to a higher plateau than optimizing all rules uniformly: pass@1 approaches $\sim 46\%$ for *CodeRule-RL* versus $\sim 45\%$ for the all-rules baseline. Auxiliary traces follow the same trend, with the normalized reward rising to $\sim 0.90$ for *CodeRule-RL* and saturating near $\sim 0.81$ for the baseline, while the KL trace remains small and stable ($< 0.006$), indicating con-

trolled updates. Since the optimizer, data, and decoding settings are identical across conditions, the improvement is attributable to the reward schedule itself: we adjust only rule weights, emphasizing the most frequently violated rules and gradually relaxing them as violation rates decline.

**Rule-wise guidance and pass@1: observational evidence.** On `Qwen2.5-Coder-3B` (Hui et al., 2024), *CodeRule-RL* improves **pass@1** even *without* unit-test prompts (20.85% vs. 20.43% base); retaining test prompts brings a modest further rise to 22.13%. However, the *CodeRule-RL* variant incorporating unit tests achieves a **pass@1** of only 18.72%. These gains are *consistent with* a mechanism in which coding-standard guidance defines *per-rule* reward shaping and a simple frequency-aware schedule concentrates weight on prevalent violations and then relaxes as they decline. Feedback is consistent and dense during RL; tests remain outside the loop; and only rule weights change while data, prompts, and the optimizer are fixed. This design reduces interference across heterogeneous rules and aligns training pressure with common failure modes, which matches the trend in our curriculum-versus-all-rules comparison (Figure 4) and the overall improvements in Table 1. We do not make a causal claim beyond these associations.

**Standard-Agnostic and cross-language generalization.** To demonstrate generalizability, we applied *CodeRule-RL* to Python under the PEP 8 standard. As detailed in Appendix J, the framework successfully transfers to this dynamic paradigm: on Qwen2.5-Coder-3B, it achieves a compliance score of **82.13%**, surpassing the strong RL baseline REAL (Yao et al., 2025) by a significant margin of **+10.28%**. Crucially, this strict enforcement implicitly safeguards functional robustness, yielding slight performance gains on LiveCodeBench (22.14%) and HumanEval. These results confirm that *CodeRule-RL* is standard-agnostic and capable of cross-language generalization, demonstrating that its efficacy extends beyond C/MISRA applications; specific details are provided in Appendix J.

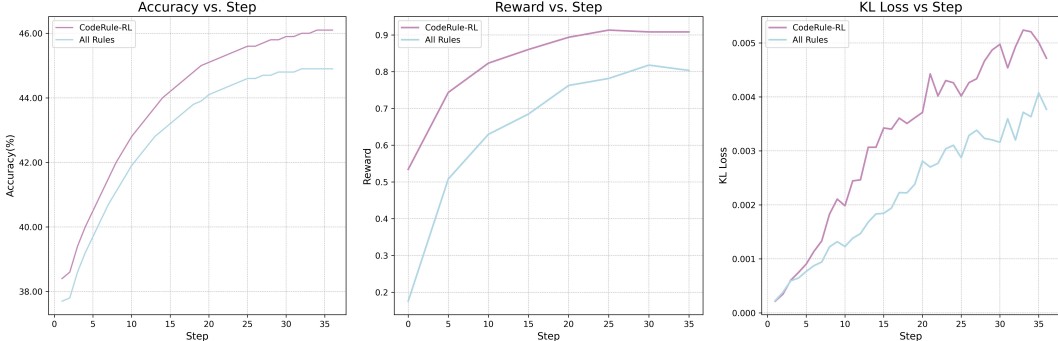

Figure 4: Training dynamics on `Qwen2.5-Coder-7B`: pass@1 (left), compliance reward (middle), and KL loss (right) over training steps for *CodeRule-RL* (curriculum) and the All-Rules baseline (no curriculum). Higher is better for pass@1 and reward; lower is better for KL loss.

## 5 CONCLUSION

Prior work has largely overlooked coding-standard guidance as auxiliary signals for training. We use this guidance only to shape per-rule rewards with the ultimate goal of improving **pass@1**. We introduce **CodeRule-RL**, a standard agnostic RL framework that maintains pass@1 as the primary evaluation metric while optimizing a rule-based proxy reward. Machine-checkable rule checks define per-rule reward shaping together with a simple frequency aware schedule implemented inside the reward. The schedule emphasizes the most frequently violated rules first and then gradually reduces their weights as violation rates decline. The optimizer, data, and prompts remain fixed; only reward weights are adjusted. Unit-test specifications may appear in prompts to express requirements, but tests are not executed during training, and we do not construct preference pairs or counterfactual negatives. On a frozen subset of CodeContests+ (C), *CodeRule-RL* attains higher **pass@1** with substantially lower training time than RL that executes tests during training, because rewards are computed from per-rule checks without running tests in the loop.

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

## A  THE USE OF LARGE LANGUAGE MODELS (LLMS)

In accordance with ICLR 2026 policies on LLM usage, we disclose that AI assistance was used only during manuscript preparation for surface-level editing. Specifically, we used ChatGPT, DeepSeek, and Grammarly to correct grammatical errors and refine wording. All ideas, claims, experiment designs, analyses, and conclusions are authored and verified by the human authors. We reviewed and edited all AI suggestions before inclusion. No confidential or under-review material, proprietary data, or private code was provided to any AI system. The authors remain fully responsible for the final content of the paper.

## B  REPRODUCIBILITY STATEMENT

All implementation and evaluation details needed for replication are specified in the paper. We document the exact dataset manifest and fixed splits, the construction of prompts and the prompt-parity *No-Test* protocol, the reward computation pipeline with analyzer versions and flags, compiler toolchains and build options for pass@1, decoding settings evaluated separately (greedy and nucleus), the definition of success and failure, and the full set of training hyperparameters including curriculum thresholds and reward clipping. We report five independent runs per setting with mean±SD, provide hardware configuration, wall-clock training time, and average reward latency, and describe known sources of non-determinism and the controls we apply. Each table and figure references the scripts and logged fields from which it is derived so results can be regenerated from the documented procedures.

## C  IMPACT OF CODING STANDARDS ON MODEL FUNCTIONALITY

Figure 5 illustrates the positive correlation between the adoption of coding standards and model performance. As training progresses, both the *CodeRule-RL* and All Rules conditions show significant improvements in functionality, with a reduction in program errors. This pattern consistently appears across different experimental setups, confirming the effectiveness of coding standards in enhancing both program quality and model performance.

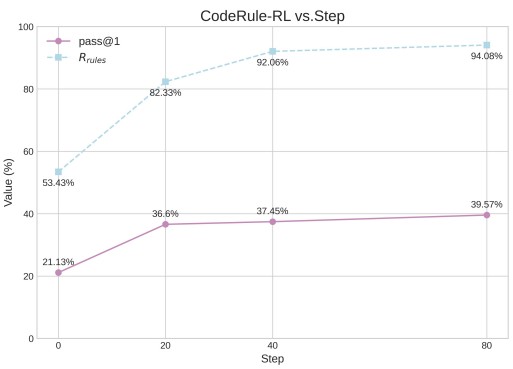 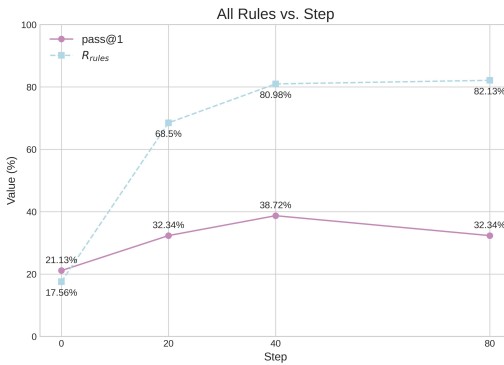

Figure 5: Correlation between functionality and training rewards. Experiments were conducted on the `Qwen2.5-Coder-7B` model. The left plot shows *CodeRule-RL*, and the right plot shows all rules.

## D  RE-EVALUATION DETAILS FOR BASELINES

All re-evaluated baselines use a unified protocol: greedy decoding ($T=0$, $k=1$) with no per-model tuning; input/output context limits identical to ours; a unified set of stop sequences (listed in the appendix); five fixed random seeds across all runs; and the same GPU class with the same compilation toolchain (GCC 13 / Clang 17).

---

**Algorithm 1** *CodeRule-RL*

---

1: **Input:**
2:    1) Initial policy $\pi_\theta$, reference policy $\pi_{\text{ref}}$.
3:    2) Number of iterations $M$.
4:    3) Static analyzer $\Psi$, rule set $\mathcal{R}$.
5:    4) Base importances $\{\bar{w}_r\}$, coverage mask $\mathbf{m}$, shaping functions $\{\psi_r\}$, caps $\{N_r\}$, gains $\{k_r\}$, EMA rate $\lambda$, mastery threshold $\tau$, window $W$, warmup $T_{\text{warm}}$, cool-down $T_{\text{cool}}$, hysteresis $h$, gate threshold $\epsilon_g$.
6:    5) GRPO group size $N$, PPO clip $\epsilon$, KL weight $\beta$.
7: **Initialize:** $\bar{\mathbf{v}}(0) \leftarrow \mathbf{0}$; $K(0) \leftarrow 1$; set $\alpha_r(0) \leftarrow 0$; clear $\{t_r^{\text{on}}, t_r^{\text{master}}\}$.
8: **for** $t = 1$ to $M$ **or** not converged **do**
9:    Sample a batch of prompts $\{q_j\}_{j=1}^B$; for each $q_j$, sample $N$ candidates $C_{j,1:N} \sim \pi_\theta(\cdot \mid q_j)$
10:    Run $\Psi$ to obtain the rule-indexed violation vector $\mathbf{v}(C_{j,k})$ for all candidates
11:    **if** analysis fails for $C_{j,k}$ **then**
12:       $R(C_{j,k}, t) \leftarrow R_{\text{fail}}$                                 ▷ hard penalty for analyzer failure
13:    **end if**
14:    **Update curriculum state:**
15:    $\bar{\mathbf{v}}(t) \leftarrow (1-\lambda)\bar{\mathbf{v}}(t-1) + \lambda \cdot \mathbb{E}_{j,k}[\mathbf{v}(C_{j,k})]$      ▷ EMA of per-rule violation frequencies
16:    Rank rules by $\bar{\mathbf{v}}(t)$; let $\mathcal{S}(t) = \text{Top-}K(t)$      ▷ active set of $K(t)$ most violated rules
17:    **if** $\forall r \in \mathcal{S}(t)$ have been under $\tau$ for $W$ steps **then**
18:       $K(t+1) \leftarrow \min\{K(t)+1, K_{\max}\}$; set $t_r^{\text{on}} \leftarrow t$ for newly activated rules
19:    **else**
20:       $K(t+1) \leftarrow K(t)$
21:    **end if**
22:    Apply hysteresis: rules previously deactivated may reenter only if $\bar{v}_r(t) \geq \tau + h$
23:    **Compute rewards:**
24:    **for** each $C_{j,k}$ **do**
25:       **for** each $r \in \mathcal{R}$ **do**
26:          $\hat{v}_r = \psi_r\big(\min\{v_r(C_{j,k}), N_r\}\big)$       ▷ cap & shape the raw violation count
27:          $P_r(\hat{v}_r) = 1 - \exp(-k_r \hat{v}_r)$            ▷ per-rule penalty in $[0, 1)$
28:          Compute schedule $\alpha_r(t)$ with warmup ($T_{\text{warm}}$), anneal/cool-down ($T_{\text{cool}}$)
29:          Gate $g_r(t, \mathbf{v}) = \mathbb{1}\{r \in \mathcal{S}(t)\} \cdot \mathbb{1}\{v_r(C_{j,k}) \geq \epsilon_g\}$      ▷ active & triggered
30:          $w_r(t, \mathbf{v}) = \bar{w}_r\, m_r\, \alpha_r(t)\, g_r(t, \mathbf{v})$       ▷ effective per-rule weight
31:       **end for**
32:       $R(C_{j,k}, t) = \text{clip}\Big(1 - \sum_{r \in \mathcal{R}} w_r(t, \mathbf{v})\, P_r(\hat{v}_r),\ -0.5,\ 1.2\Big)$    ▷ static-analysis-only reward
33:    **end for**
34:    **Optimize the policy $\pi_\theta$:**
35:    For each group $j$, center advantages $A_{j,k} = R(C_{j,k}, t) - \frac{1}{N}\sum_{u=1}^N R(C_{j,u}, t)$
36:    Update $\theta$ with PPO-style clipped objective (clip $\epsilon$), KL penalty $\beta\,\text{KL}(\pi_\theta \| \pi_{\text{ref}})$, and a small entropy bonus
37: **end for**
38: **Output:** Trained generator $\pi_\theta$.

---

# E   Vulnerability detection

Traditional code reviews and dynamic testing often fail to cover all edge cases, leaving potential issues undetected. To provide a more comprehensive evaluation of the generated code, we employ **Infer**, a state-of-the-art static analysis tool. Unlike dynamic testing, which executes the code, Infer conducts an in-depth analysis of the source code to identify potential vulnerabilities and runtime errors that are challenging to detect through conventional methods.

For instance, Infer can track the complete lifecycle of variables, enabling it to flag memory management errors such as improper memory allocation and deallocation. Furthermore, it identifies performance bottlenecks like **EXPENSIVE_LOOP_INVARIANT_CALL**. This issue occurs when a computationally expensive function (determined through cost analysis to have at least linear complexity) that is loop-invariant is called inside a loop. This inefficient coding pattern can severely degrade performance, especially when the code is executed repeatedly.

To quantify the security and robustness of the generated code, we introduce the **Vulnerability-Free Rate (VFR)** metric. This metric measures the percentage of code samples that pass the Infer static

Table 3: Performance Comparison of Baseline and Optimized Models (Pass@1 and VFR for Vulnerability Detection)

| Model | pass@1 (%) | VFR(%) |
|-------|-----------|--------|
| Absolute_Zero_Reasoner-Coder-3b | 15.74 | 77.87 |
| NextCoder-7B | 36.60 | 87.02 |
| Qwen3-4B-Instruct-2507 | 55.32 | 77.02 |
| Qwen3-4B-Instruct-2507 w / *CodeRule-RL* | 56.17 (**+0.85**) | 80.43 (**+3.41**) |
| Deepseek-Coder-1.3B | 2.13 | 41.70 |
| Deepseek-Coder-1.3B w / *CodeRule-RL* | 6.00 (**+3.87**) | 77.02 (**+35.32**) |
| Deepseek-Coder-6.7B | 18.72 | 89.36 |
| Deepseek-Coder-6.7B w / *CodeRule-RL* | 28.09 (**+9.37**) | 90.64 (**+1.28**) |
| Qwen2.5-Coder-1.5B | 2.55 | 91.91 |
| Qwen2.5-Coder-1.5B w / *CodeRule-RL* | 11.49 (**+8.94**) | 88.47 (**-3.44**) |
| Qwen2.5-Coder-3B | 20.43 | 83.02 |
| Qwen2.5-Coder-3B w / *CodeRule-RL* | 22.13 (**+1.70**) | 88.94 (**+5.92**) |
| Qwen2.5-Coder-7B | 21.13 | 82.02 |
| Qwen2.5-Coder-7B w / *CodeRule-RL* | 39.57 (**+18.44**) | 84.68 (**+2.66**) |

analysis scan without any detected issues. It is mathematically defined as:

$$\text{VFR} = \frac{1}{|\mathcal{X}_{\text{eval}}|} \sum_{x \in \mathcal{X}_{\text{eval}}} \nVdash[F_{BI}(x) = 1].$$ (16)

where, $F_{BI}$ is Facebook Infer vulnerability detector. The VFR complements the **pass@1** metric, which measures functional correctness, to form a comprehensive framework for evaluating model performance.

As shown in Table 3, our experiments reveal that the optimization method generally improves the functional correctness (**pass@1**) of the models. However, its impact on code security (**VFR**) varies across different models. For example, after optimization, `Qwen2.5-Coder-1.5B` shows a **+8.94%** improvement in **pass@1** but a **-3.44%** decrease in its VFR. This suggests that while the new code is more functionally correct, it also introduces more potential issues detectable by static analysis. In contrast, larger models like `Qwen2.5-Coder-7B` demonstrate excellent performance on both fronts, achieving a significant **+18.44%** increase in **pass@1** alongside a solid **+2.66%** improvement in VFR. This result indicates that our optimization method, when applied to larger models, can effectively enhance problem-solving capabilities without sacrificing code quality or security.

## F DATA PROCESSING PIPELINE

The pipeline in Figure 6 follows Sec. 3.1 and prepares the data used for training. Tasks are mined and decontaminated, then pass through up to three *probe-and-check* iterations per prompt. Probe candidates that fail to compile are discarded. A rule evaluator $\Phi$ produces *per rule* signals $\mathbf{s}(C)$ that are used *only* for offline screening and aggregate quality checks; these signals are *not* stored as labels and do *not* become part of the training corpus. $D_{\text{ID}}$ and $D_{\text{RM}}$ are temporary inspection sets for reporting and analysis, not supervision. The final frozen training set $\mathcal{D}$ contains prompts and minimal metadata. During RL, per rule signals are recomputed online by $\Phi$ on the sampled candidate and are used only inside the reward (Secs. 3.3–3.4); data, prompts, and the optimizer remain fixed.

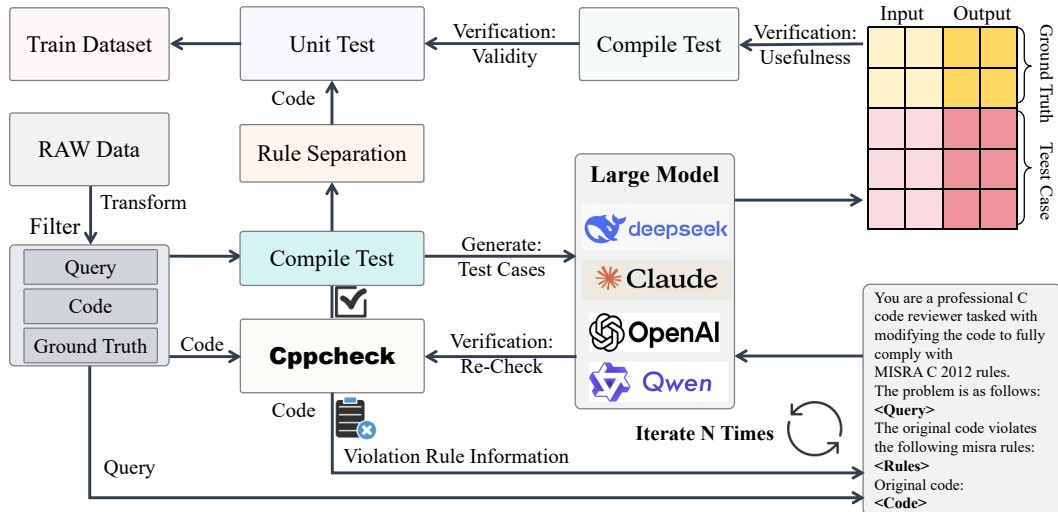

Figure 6: **Data pipeline.** Tasks are mined and decontaminated, followed by up to three probe-and-check iterations for curation. The evaluator Φ produces per rule signals for filtering and diagnostics during curation; these signals are discarded after curation and are not persisted in the dataset. During RL, signals are recomputed online for reward shaping. The pipeline is standard-agnostic: replacing the rule set or checker changes only Φ and its mapping, not the learning algorithm.

## G   ABLATION STUDY FOR COMPLIANCE

**Coding-standard compliance.** We assess MISRA C:2012 compliance using `cppcheck` 2.7 with the MISRA addon. With one candidate per prompt ($k=1$), we report *compliance*@1, *pass*@1 and *joint*@1. Advisory findings are logged but ignored by the metric. Unless otherwise stated, each translation unit is analyzed with `cppcheck` 2.7.

**Analysis.**   To evaluate the impact of training regimes and rule granularity, we benchmark *CodeRule-RL* against three baselines (Figure 7). Optimizing *All Rules* simultaneously fails to induce rule adherence (joint@1 = 0) due to severe gradient interference. The *Single Rule* approach achieves high compliance peaks but suffers from catastrophic forgetting: at Step 60, despite 57.02% compliance, pass@1 collapses from 39.15% to 17.45%, as strict isolation destroys general coding capability. *Mixed Rules* eventually converges to competitive performance (24.68% joint@1) but exhibits a cold start phenomenon, lagging significantly behind *CodeRule-RL* in the early phase (8.51% vs. 12.77% joint@1 at Step 20). *CodeRule-RL* demonstrates the optimal trade-off between efficiency and stability. A phased curriculum accelerates early learning compared to *Mixed Rules*, maintaining robust functional correctness (38.30%) without the degradation seen in *Single Rule*. These results support our mechanism: per rule credit assignment and gradient isolation within the active frontier drive compliance and joint success, whereas optimizing all rules simultaneously induces gradient interference and stalls compliance learning.

### G.1   HUMANEVAL AND MBPP

For our evaluation on general benchmarks, we used **HumanEval**Chen et al. (2021), **HumanEval Plus**, **MBPP**Austin et al. (2021), and **MBPP Plus** to assess Python programming tasks on EvalPlus Liu et al. (2023b). The HumanEval Plus dataset extends the original HumanEval test cases by a factor of 80 to create the HumanEval Plus dataset, while MBPP Plus includes 35 times more test cases than the original MBPP.

Table 4 summarizes the results on HumanEval (and HumanEval Plus) Chen et al. (2021) and MBPP (and MBPP Plus) Austin et al. (2021). After applying our optimization, the 1.5B and 3B models demonstrate substantial improvements across all metrics. Notably, the 1.5B model shows an over 16% increase in its HE metric, indicating that our method effectively enhances the code-generation capabilities of smaller models. In contrast, the 4B and 7B models exhibit only marginal improve-

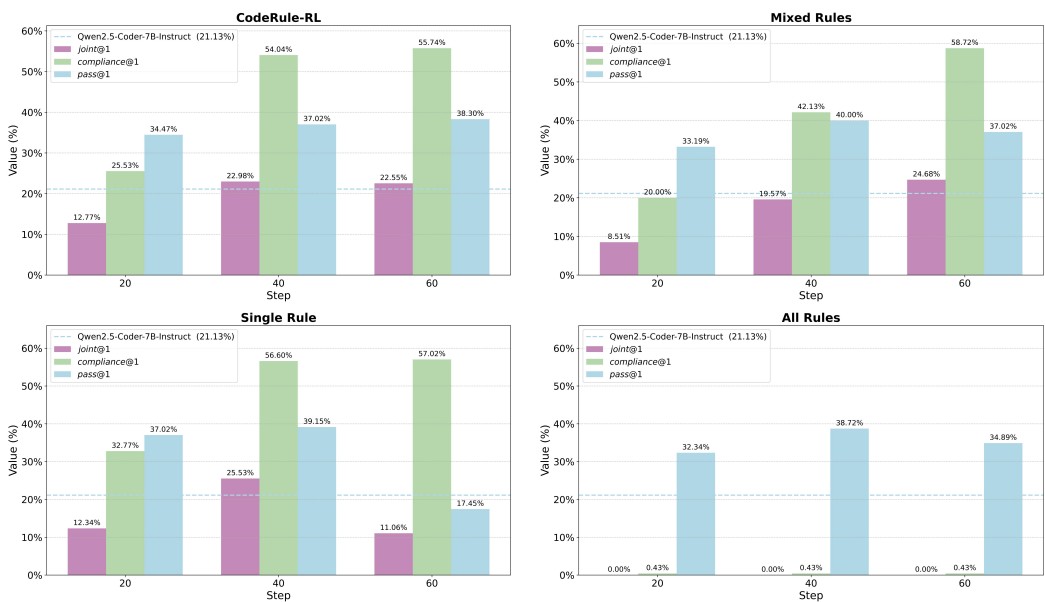

Figure 7: Qwen2.5–Coder–7B under different training regimes and rule granularities. *CodeRule-RL*: a frequency-driven, phased curriculum that learns multiple rules jointly. **Mixed Rules**: incorporates only the required rules without curriculum scheduling. **Single Rule**: focuses on optimizing individual rules independently. **All Rules**: optimizes the full rule set simultaneously.

ments, likely because these larger models are already approaching the performance ceiling on these benchmarks. Overall, the results suggest that our method incurs little to no degradation in general-purpose coding performance and may even result in slight gains.

Table 4: Performance Comparison of Models on Code Generation Tasks (HumanEval/MBPP, All Metrics in %)

| Model | HumanEval | | MBPP | |
| --- | --- | --- | --- | --- |
| | HE | HE+ | MBPP | MBPP+ |
| **BASE** | | | | |
| Qwen2.5-Coder-1.5B | 59.10 | 53.70 | 68.50 | 58.50 |
| Qwen2.5-Coder-3B | 76.20 | 68.90 | 70.40 | 59.30 |
| Qwen3-4B-Instruct-2507 | 76.80 | 70.10 | 80.70 | 68.30 |
| Qwen2.5-Coder-7B | 81.70 | 75.60 | 82.30 | 68.50 |
| Deepseek-coder-1.3B | 61.60 | 57.90 | 63.80 | 54.50 |
| Deepseek-coder-6.7B | 68.90 | 63.40 | 76.20 | 64.80 |
| **Ours** | | | | |
| Qwen2.5-Coder-1.5B | 75.60 | 68.90 | 70.10 | 59.50 |
| Qwen2.5-Coder-3B | 74.40 | 67.70 | 71.20 | 60.30 |
| Qwen3-4B-Instruct-2507 | 77.40 | 72.00 | 80.70 | 68.30 |
| Qwen2.5-Coder-7B | 81.70 | 76.20 | 83.30 | 68.30 |
| Deepseek-coder-1.3B | 60.40 | 57.30 | 65.10 | 56.10 |
| Deepseek-coder-6.7B | 69.50 | 62.80 | 75.70 | 64.00 |

## H  THE PARAMETER $\lambda$ IN FREQUENCY-DRIVEN CURRICULUM

Table 5 illustrates the sensitivity of the frequency-driven curriculum to the parameter $\lambda$. When $\lambda = 0$, the model disregards historical violation frequencies; while it maintains a reasonable pass rate, both compliance and joint scores remain at zero. Small values of $\lambda$ (e.g., 0.1, 0.2) yield only marginal performance gains. However, setting $\lambda = 0.3$ triggers a substantial performance leap, achieving a global optimum with a joint@1 of 22.26% and a compliance@1 of 55.74%. Conversely, increasing $\lambda$ to 0.5 causes a sharp collapse in performance, likely due to the model becoming overly sensitive to noise in the coding norms of historical batches.

Table 5: Influence of Frequency-Driven Curriculum $\lambda$.

| $\lambda$ | joint@1 (%) | compliance@1 (%) | pass@1 (%) |
|---|---|---|---|
| Qwen2.5-Coder-7B-Instruct | 0.00 | 0.00 | 21.13 |
| 0.10 | 6.38 | 65.11 | 27.23 |
| 0.20 | 10.21 | 59.57 | 24.26 |
| 0.30 | 22.26 | 55.74 | 39.57 |
| 0.50 | 13.62 | 32.77 | 37.45 |

## I  EFFORT OF INTERVAL WEIGHTS $\tau$

We conducted experiments on the initial weights of the sum of major violations and other minor violations at each stage. As shown in Table 6, our research found that major violation rules cannot be completely ignored, nor can they be insufficiently enforced. If the initial weight is too large, it will lead to excessive punishment during the training process, and directly reaching the reward boundary will affect the model's learning; if the weight is too small, the reward mechanism will rarely be triggered if the generated program mainly follows the major norms, but the secondary rules will not impose punishment. We conducted experiments on the initial weights of the sum of major violations and other minor violations at each stage. As shown in Table 6, our research found that major violation rules cannot be completely ignored, nor can they be insufficiently enforced. If the initial weight for major violations is too large, it will lead to excessive punishment during the training process, and directly reaching the reward boundary will disrupt the model's learning, as it may cause the model to focus too much on avoiding major violations at the expense of other important behaviors. On the other hand, if the weight for major violations is too small, the reward mechanism will rarely be triggered when the generated program mainly adheres to major norms, and the model will not be penalized for ignoring major violations. This imbalance could result in a failure to properly enforce the primary coding standards, while secondary rules fail to have enough impact. The optimal weight configuration lies in striking a balance, ensuring that major violations are sufficiently enforced without overwhelming the model, and that the secondary rules can still provide appropriate guidance.

Table 6: Influence of Interval Weights $\tau$

| $\tau$ | join@1 (%) | compliance@1 (%) | pass@1 (%) |
|---|---|---|---|
| Qwen2.5-Coder-7B-Instruct | 0.00 | 0.00 | 21.13 |
| 0.00 | 0.00 | 1.28 | 32.77 |
| 0.05 | 0.00 | 0.85 | 35.74 |
| 0.10 | 24.26 | 55.74 | 39.57 |
| 0.20 | 2.13 | 6.38 | 25.11 |
| 0.50 | 1.70 | 11.06 | 21.70 |

## J  ANNEALING WARMUP EPOCH $T_{warm}$

We use the Qwen2.5-Coder-7B model to compare the impact of different annealing steps on coding standards and functionality. The step size of the annealing process plays a crucial role in learning

major violation rules. A larger step size leads to a smaller effective weight for minor rules, thereby prioritizing the learning of coding standards related to major violations. As shown in Table 7, we selected a step size of 30 for our experiments, as it demonstrated the most significant impact on key performance metrics. Considering that Decay and Warmup are mutually corresponding processes and exhibit logical symmetry, the value of the cooling steps ($T_{cool}$) is set to be consistent with that of the warmup steps ($T_{warm}$).

Table 7: Influence of Annealing Warmup Epoch $T_{warm}$

| Epoch | join@1 (%) | compliance@1 (%) | pass@1 (%) |
|---|---|---|---|
| 0 | 0.00 | 0.00 | 21.13 |
| 5 | 8.09 | 12.34 | 35.74 |
| 10 | 7.66 | 21.70 | 24.47 |
| 20 | 4.26 | 6.81 | 36.60 |
| 30 | 22.26 | 55.74 | 39.57 |
| 50 | 4.26 | 8.51 | 35.32 |

## K  THE IMPACT OF DIFFERENT TOP-K VALUES

We adopted the parameters consistent with those in Section 4, used Qwen2.5-Coder-7B as the base model, set the total number of steps to 20, and adjusted different Top-K values. As shown in Table 8, each time a different $K$ is selected, the model's coding standard violation rate increases as the number of selections increases. Although the Pass@1 score when $K = 5$ is slightly higher than that when $K = 1$, we still chose $K = 1$ after comprehensive consideration. Theoretically, if $K$ approaches infinity, it would be an experiment involving all rules, and the results can be referred to in Section G.

Table 8: Impact of Different Top-k Values $K$

| Model | violated(%) | pass@1(%) |
|---|---|---|
| Qwen2.5-Coder-7B-Instruct | 99.57 | 21.13 |
| *CodeRule-RL w/K=1* | 85.11 | 34.47 |
| *CodeRule-RL w/K=2* | 91.91 | 34.02 |
| *CodeRule-RL w/K=5* | 98.72 | 35.74 |

## L  EXPERIMENTS ON PYTHON PEP8 GENERALIZATION

**Dataset**: We followed the same datasets pipline as with the C-language training dataset, as shown in Figure 6. First, we filtered 5,000 samples from BigCode rStarCoder's `seed_testcase`[1]. Based on the analysis results from `pylint`, `mypy`, and `ruff` on the model-generated code, we incorporated PEP8 guidelines and related rule samples into the prompt. **Model**: We selected Qwen2.5-Coder-3B and Qwen2.5-Coder-7B as the base models for our experiments. **Baseline**: For the unit testing and code iteration optimization framework CURE, we directly used the original weights provided in Wang et al. (2025a) for evaluation. To ensure consistency in our implementation, we made the following adjustments of REALYao et al. (2025): we replaced the original `mypy`-only check with three tools—`pylint`, `mypy`, and `ruff`—to analyze Python code from different angles. This reflects Python's advantage over C-language in static analysis. Additionally, to align with CURE's implementation, we increased the maximum number of unit tests from 5 to 16. The REAL experimental settings were as follows: total training steps of 800, mini batch size of 256, micro batch size

---

[1] https://huggingface.co/datasets/microsoft/rStar-Coder

of 4, with static analysis and unit testing weights set to 0.50 each. **Evaluation**: We used `EvalPlus` to evaluate the `HumanEval`, `HumanEvalPlus`, `MBPP`, `MBPPPlus`, and `LiveCodeBench V6` datasets. For static analysis, instead of creating a separate validation set, we generated samples using `LiveCodeBench` and performed `pylint`, `mypy`, and `ruff` code style checks. The scores were primarily based on `pylint`, and for each issue, we averaged the scores from 10 generated code samples, which we refer to as the compliance score.

As shown in Table 9 and Table 10, this confirms the exceptional capability of **CodeRule-RL** in adhering to PEP 8 standards. *CodeRule-RL*-7B significantly improves compliance to **77.08%**, outperforming both the baseline Qwen2.5-Coder-7B (73.40%) and ReasonFlux-Coder-7B (73.58%); meanwhile, at a smaller parameter scale, *CodeRule-RL* demonstrates superior rule injection efficacy, with **CodeRule-RL-3B** achieving the highest compliance score of **82.13%** across all models, drastically surpassing its size-matched baseline (70.51%) and the 7B variants, proving the effectiveness of dense rule-based feedback in mastering fine-grained standards. This strict compliance does not compromise functional correctness, as *CodeRule-RL*-7B outperforms the baseline on both MBPP (83.60%) and LiveCodeBench (27.31%), achieving an optimal balance between high compliance and functional capability.

Table 9: Performance of various instruct models on HumanEval($^+$) and MBPP($^+$).

| Model | compliance@1 | HumanEval | | MBPP | |
|---|---|---|---|---|---|
| | | HE | HE+ | MBPP | MBPP+ |
| Qwen2.5-Coder-7B | 73.40 | 81.70 | 75.60 | 82.30 | 68.50 |
| ReasonFlux-Coder-7B | 73.58 | 69.50 | 62.20 | 78.60 | 64.30 |
| *CodeRule-RL*-7B | **77.08** | 81.70 | 75.60 | **83.60** | **70.10** |
| Qwen2.5-Coder-3B | 70.51 | 76.20 | 68.90 | 70.40 | 59.30 |
| REAL-3B | 71.85 | 77.40 | 70.10 | **72.20** | **60.60** |
| *CodeRule-RL*-3B | **82.13** | **78.44** | **69.71** | 71.93 | 59.95 |

Table 10: Performance of various instruct models on LiveCodeBench V6(2305-2504).

| Model | compliance@1 | Livecodebench | | |
|---|---|---|---|---|
| | | pass@1 | pass@5 | pass@10 |
| Qwen2.5-Coder-7B | 73.40 | 26.69 | 31.53 | 33.36 |
| ReasonFlux-Coder-7B | 73.58 | 23.60 | 28.67 | 30.43 |
| *CodeRule-RL*-7B | **77.08** | **27.31** | **31.99** | **33.55** |
| Qwen2.5-Coder-3B | 70.51 | 20.70 | 26.88 | **29.00** |
| REAL-3B | 71.85 | 20.53 | 25.89 | 27.49 |
| *CodeRule-RL*-3B | **82.13** | **22.14** | **27.22** | 28.63 |

## M    RELATIONSHIP BETWEEN COMPLIANCE WITH CODING STANDARDS AND PASS@1

We conduct experimental analysis on three models: Qwen2.5-Coder-7B, and DeepSeek-Coder-6.7B. The experimental parameters are consistent with those in Section 4, and we perform Cppcheck and CodeContents+ evaluations every 20 steps. In this context, $Violated = 1 - compliance@1$ represents the proportion of generated code that violates coding standards. We also present the number of violations of mandatory and recommended coding standards during the training process. As shown in Figure 8, with the progress of training, the coding standard violation rate exhibits negative growth, while the functional indicators show positive growth. This demonstrates a balanced relationship between compliance with coding standards and functionality: the more the generated code adheres to coding standards, the higher the Pass@1 rate of the generated code. As illustrated in Figure9, 11, and 10, the number of violations of required and advisory rules gradually decreases as training proceeds. The reduction in the required rule violation rate is greater than that of the advisory rule violation rate. It can be observed that the required rule violation rate decreases significantly in

the interval of 20 to 40, which is related to our setting of $T_{\mathrm{Warm}}$ as 30. Additionally, we have counted the number of violations of MISRA C:2012 Rule 8.2, which has the highest number of violations in 12. With the progress of training, we have controlled the violation rate to almost 0%, and different models show a basically consistent trend.

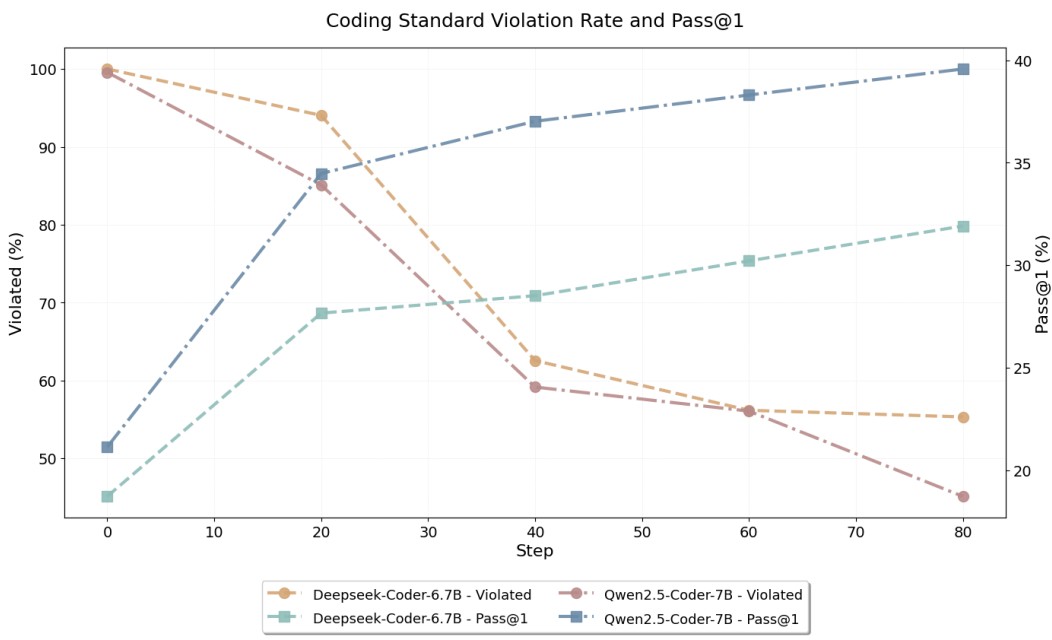

Figure 8: The Relationship Between Compliance with Coding Standards and Pass@1.

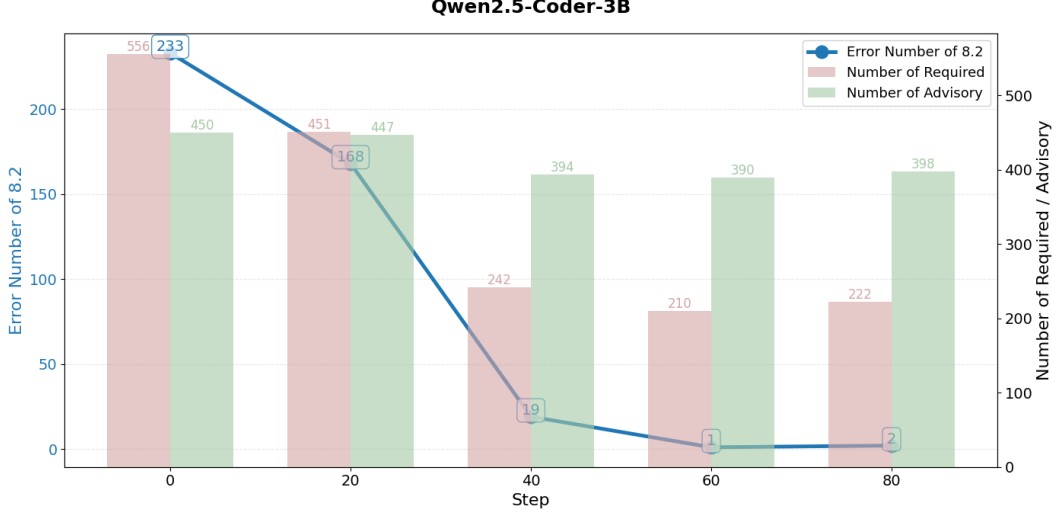

Figure 9: Performance of Required and Advisory Coding Standard Violations During Qwen2.5-Coder-3B.

## N  LEARNING SEQUENCE

After the second iteration of the data generation phase (as shown in Figure 6), we selected 4,000 compilable code samples for code style rule violations detection. Based on these samples, we evalu-

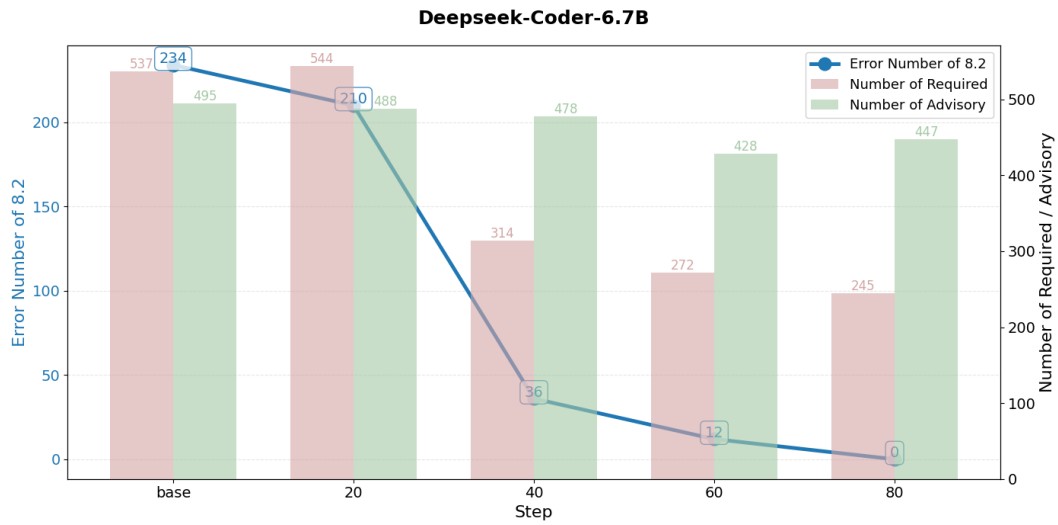

Figure 10: Performance of Required and Advisory Coding Standard Violations During DeepSeek-Coder-6.7B.

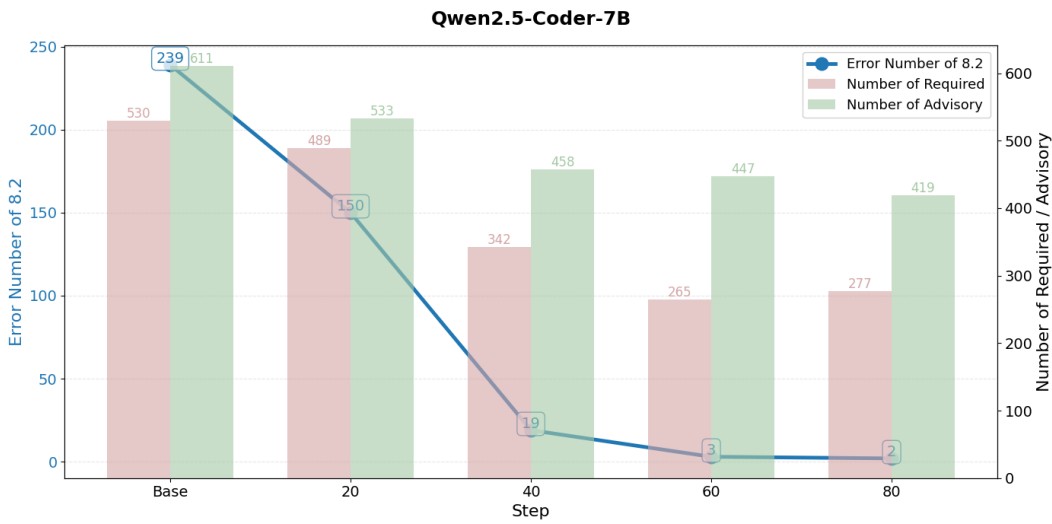

Figure 11: Performance of Required and Advisory Coding Standard Violations During Qwen2.5-Coder-7B.

ated the model's error rate for different coding rules, with the results shown in Figure 12. The figure illustrates the number of violations for various MISRA C:2012 rules, with `misra-c2012-8.2` having the highest number of violations, reaching 4,473, followed by `misra-c2012-15.6` with 2,240 violations. Most other rules had relatively few violations, reflecting the varying levels of adherence to different rules in coding. Using this violation data, we followed the corresponding sequence for training, detection, and evaluation to fine-tune the model's performance more effectively.

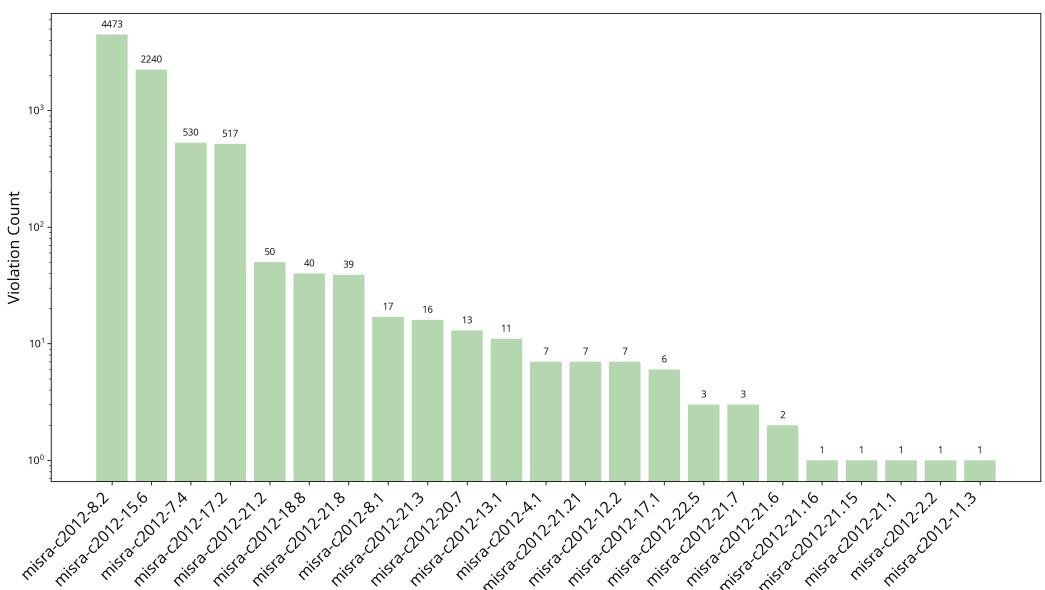

Figure 12: MISRA C:2012 Rule Violation Distribution.

Table 11: MISRA C:2012 rule sets for each category: Mandatory, Required, and Advisory. Counts reflect analyzer outputs, including duplicates. All counts are in percentage.

| Category | Rules (IDs) | Count |
|----------|-------------|-------|
| Mandatory | 7.5, 9.1, 12.5, 17.3, 17.4, 17.6, 17.9, 19.1, 21.13, 21.17, 21.18, 21.19, 21.2, 21.22, 22.2, 22.4, 22.5, 22.6 | 18 |
| Required | 1.1, 1.3, 1.4, 1.5, 2.1, 2.2, 3.1, 3.2, 4.1, 5.1, 5.2, 5.3, 5.4, 5.5, 5.6, 5.7, 5.8, 6.1, 6.2, 6.3, 7.1, 7.2, 7.3, 7.4, 8.1, 8.2, 8.3, 8.4, 8.5, 8.6, 8.8, 8.12, 8.14, 8.15, 9.2, 9.3, 9.4, 9.5, 10.1, 10.2, 10.3, 10.4, 10.6, 10.7, 10.8, 11.1, 11.2, 11.3, 11.6, 11.7, 11.8, 11.9, 12.2, 13.1, 13.2, 13.5, 13.6, 14.1, 14.2, 14.3, 14.4, 15.2, 15.3, 15.6, 15.7, 16.1, 16.2, 16.3, 16.4, 16.5, 16.6, 16.7, 17.1, 17.2, 17.5, 17.13, 18.1, 18.2, 18.3, 18.6, 18.7, 18.8, 18.9, 20.2, 20.3, 20.4, 20.6, 20.7, 20.8, 20.9, 20.11, 20.12, 20.13, 20.14, 21.1, 21.2, 21.3, 21.4, 21.5, 21.7, 21.8, 21.9, 21.12, 21.14, 21.15, 21.16, 21.21, 21.23, 21.24, 22.1, 22.3, 22.7, 22.8, 22.9, 22.11, 22.15, 22.16, 22.17, 23.2, 23.4, 23.6, 23.8 | 122 |
| Advisory | 1.2, 2.3, 2.4, 2.5, 2.6, 2.7, 4.2, 5.9, 8.7, 8.9, 8.11, 8.13, 8.16, 8.17, 10.5, 11.4, 11.5, 12.1, 12.3, 12.4, 13.3, 13.4, 15.1, 15.4, 15.5, 17.8, 17.11, 17.12, 18.4, 18.5, 19.2, 20.1, 20.5, 21.11, 23.1, 23.3, 23.5, 23.7 | 38 |

## O  CODING STANDARDS

As shown in Table 11, we adopt the 178 rules of the MISRA C:2012 standard as the coding guideline (including 18 Mandatory, 122 Required, and 38 Advisory rules) and use the `cppcheck` tool for analysis. Because Advisory rules have minimal impact on coding in software engineering projects,

we perform training, detection, and evaluation for the S metric only on the Mandatory and Required categories. Furthermore, since the study uses single-file C data, MISRA C:2012 Rule 21.6 (which prohibits the use of standard library input/output functions in `<stdio.h>` and `<wchar.h>`, such as `printf` and `fgets`) would render `stdio.h` unusable and hinder subsequent file read/write operations needed for evaluation; therefore, this rule was excluded from the process. Ultimately, the rules actually included in the evaluation comprise 18 Mandatory and 121 Required rules, for a total of 139 rules.

## P  PERFORMANCE OF INSTRUCT MODELS ON MULTIPL-E

Beyond our constructed Contents+ benchmark, we further evaluated the code generation capabilities of instruction-tuned models in C and C++ contexts. MultiPL-E Cassano et al. (2023) extends the original HumanEval and MBPP benchmarks by translating Python tasks and unit tests into 22 other programming languages (including C++, Java, PHP, TypeScript, C#, Bash, and JavaScript). Evaluating on MultiPL-E provides a comprehensive view of model performance across diverse languages and helps disentangle language-specific factors affecting LLM code generation. We specifically selected the C and C++ subsets for this assessment, The batch_size for testing is 20, the temperature is 0.2, and the completion_limit is 20, with results presented in Table 12. *CodeRule-RL* slightly outperforms base instruction models on the C/C++ versions of HumanEval and MBPP, demonstrating the effectiveness and generalization capability of our approach.

Table 12: Pass@1 (%) on MultiPL-E benchmarks. Comparison between Qwen2.5-Coder-3B-Instruct and the *CodeRule-RL* variant.

| Model | HumanEval | | MBPP | |
|---|---|---|---|---|
| | **Python** | **C++** | **Python** | **C ++** |
| Qwen2.5-Coder-3B | 76.20 | 62.76 | 70.40 | 56.13 |
| *CodeRule-RL* | 74.40(-1.80) | 62.80(+0.04) | 71.20(+0.80) | 58.40(+2.27) |

Table 13: Comparison of base models and *CodeRule-RL* variants on compliance@1, pass@1, and joint@1. Report mean±SD across trials.

| Model | join@1 (%) | compliance@1 (%) | pass@1 (%) |
|---|---|---|---|
| AZR-Coder-3b (Zhao et al., 2025) | 0.00 | $0.85 \pm 0.21$ | $15.74 \pm 0.65$ |
| NextCoder-7B (Aggarwal et al., 2025) | 0.00 | $1.70 \pm 0.28$ | $36.60 \pm 0.82$ |
| Seed-Coder-8B (2025) | 0.00 | $0.85 \pm 0.18$ | $37.45 \pm 0.86$ |
| Deepseek-Coder-1.3B (2024) | 0.00 | $0.85 \pm 0.21$ | $2.13 \pm 0.31$ |
| Deepseek-Coder-1.3B w / *CodeRule-RL* | $2.64 \pm 0.35$ | $2.55 \pm 0.33$ | $6.00 \pm 0.48$ |
| Deepseek-Coder-6.7B (2024) | 0.00 | 0.00 | $18.72 \pm 0.71$ |
| Deepseek-Coder-6.7B w / *CodeRule-RL* | $11.91 \pm 0.58$ | $36.17 \pm 0.89$ | $28.09 \pm 0.79$ |
| Qwen2.5-Coder-1.5B (2024) | 0.00 | $0.43 \pm 0.14$ | $2.55 \pm 0.34$ |
| Qwen2.5-Coder-1.5B w / *CodeRule-RL* | $4.26 \pm 0.42$ | $41.28 \pm 0.94$ | $11.49 \pm 0.55$ |
| Qwen2.5-Coder-3B (2024) | 0.00 | $0.43 \pm 0.18$ | $20.43 \pm 8.61$ |
| Qwen2.5-Coder-3B w / *CodeRule-RL* | $9.80 \pm 0.51$ | $17.45 \pm 0.66$ | $22.13 \pm 2.76$ |
| Qwen2.5-Coder-7B (2024) | 0.00 | 0.00 | $21.13 \pm 0.75$ |
| Qwen2.5-Coder-7B w / *CodeRule-RL* | $\mathbf{24.26 \pm 0.81}$ | $\mathbf{55.74 \pm 1.05}$ | $\mathbf{39.57 \pm 0.92}$ |

# Q    PROMPT OF CURE AND SETTING OF UNIT TEST

To calculate rewards via post-generation unit testing and mitigate the scarcity of valid unit tests, we utilized the external DeepSeek API to generate supplementary test cases, bringing the total to 16 tests per task (matching the test suite size of CURE Wang et al. (2025a)). This approach of combining *CodeRule-RL* with unit tests aligns with Real Yao et al. (2025). We assigned equal weights of 0.5 to both coding standard compliance and unit test results. The prompts used for code generation in CURE and for unit test generation are shown in Figure 13 and Figure 14, respectively.

<|im_start|>**System:**

You are a helpful assistant help user solve problems.

<|im_end|>

<|im_start|>**User:**

You need to think first then write {{language}} script. {{special_requirements}}.

If the language is C or C++, the generated program must comply with MISRA C 2012,

which states that "All declarations of an object or function shall use the same name and type".

This is the problem:

{{problem}}

<|im_end|>

<|im_start|>**Assistant:**

Figure 13: Prompt of Using the CURE FrameworkWang et al. (2025a).

<|im_start|> **System:**
You are a helpful assistant help user generate test examples for coding tasks.
<|im_end|>
<|im_start|>**User:**
Given a coding task, instead of providing the final script, your task is to generate a new test example (both input, output and explanation).
This is the problem:
{{problem}}
{{example_intro}}
You need to provide a new test example. A good test example should be completely accurate and conform to the problem's format requirements, while also possessing enough discriminative power to distinguish correct code from incorrect code.
Before providing a test example, you must think carefully and reason step by step to derive an input and output you are very confident are correct. For example, start by designing an input you can reliably handle, then compute the output step by step. If you're unsure about the output, revise or re-design the input to ensure accuracy. Directly providing input/output pairs without this process is discouraged, as it often results in low accuracy.
Finally, after completing these previous thinking and derivation steps (you should not write the final test example unless you have gone through these steps very thoroughly), you MUST put your final test example in the following format:
**Test Input:**
```input here```
**Test Output:**
```output here```
**Explanation:**
explanation here.
 <|im_end|>
<|im_start|>**Assistant:**

Figure 14: Generate Unit Test Prompt of Using the CURE FrameworkWang et al. (2025a).

## R    CASE EXAMPLES: PASS@1 PERFORMANCE IMPROVES AS COMPLIANCE INCREASES.

As shown in Figs. 16, 3, and 15, we present an illustrative comparison between `Qwen2.5-Coder-7B-Instruct` and our trained model. The three panels respectively show the input prompt, the generated C program, and the `cppcheck` diagnostics. We evaluate on the subset of CodeContests+ tasks, and the evaluation pipeline is fully automated with no manual intervention or post-processing. **Functional correctness.** On the illustrated example, our model passes all 34 CodeContests+ unit tests for the task, whereas the `Qwen2.5-Coder-7B-Instruct` baseline passes 19. These counts refer to functional test cases on CodeContests+ and are independent of the static-analysis diagnostics in Figure 15. **Static-analysis diagnostics.** Independently of functional testing, we run `cppcheck` and summarize *rule indexed* findings under the analyzer's MISRA C:2012 configuration. Most findings for both models fall under the Required category. Under our accounting (counting Mandatory+Required and excluding Rule 21.6), and without any manual edits, our model reduces the average number of flagged issues by roughly $2\times$ relative to the baseline. Fig. 15 provides the per-rule breakdown.

## Qwen2.5-Coder-7B-Instruct

---

**test.c:4:9: style: Function types shall be in prototype form with named parameters**    [misra-c2012-8.2]
int main() {
        ^

**test.c:20:35: style: The precedence of operators within expressions should be made explicit**    [misra-c2012-12.1]
        if (current_digit + 1 >= 10) {
                          ^

**test.c:6:10: style: The value returned by a function having non-void return type shall be used**  [misra-c2012-17.7]
    scanf("%s", input);
       ^

**test.c:31:15: style: The value returned by a function having non-void return type shall be used**  [misra-c2012-17.7]
        printf("%s", input);
             ^

**test.c:33:15: style: The value returned by a function having non-void return type shall be used**  [misra-c2012-17.7]
        printf("GOTO Vasilisa.");
             ^

**test.c:1:0: style: The Standard Library input/output functions shall not be used**       [misra-c2012-21.6]
#include <stdio.h>
^

---

## Ours

---

**test.c:19:25: style: The precedence of operators within expressions should be made explicit**    [misra-c2012-12.1]
    if (last_digit != 9 && next_digit < 5) {
                    ^

**test.c:1:0: style: The Standard Library input/output functions shall not be used**       [misra-c2012-21.6]
#include <stdio.h>
^

---

Figure 15: Cppcheck diagnostics for the candidate program.

## S    CASE EXAMPLES: INCREASED MISRA C COMPLIANCE WITH COMPROMISED PASS@1 PERFORMANCE

As illustrated in Figure 17, we observe that enhanced coding standards may alter the implementation logic of the original code. The original instruction model successfully passed all 31 test cases, whereas the post-trained code failed every single test. Enforcing compliance with coding standards can, to a certain extent, impact the model's handling of code implementation logic. However, generating high-quality code is a prerequisite for engineering development, which underscores the crucial role of the foundation model's capabilities.

<|im_start|>**system**
You are Qwen, created by Alibaba Cloud. You are a helpful assistant.
<|im_end|>
<|im_start|>**user**
Please generate a C program that solves the following programming problem:
Mad scientist Mike entertains himself by arranging rows of dominoes. He doesn\'t need dominoes, though: he uses rectangular magnets instead. Each magnet has two poles, positive (a "plus") and negative (a "minus"). If two magnets are put together at a close distance, then the like poles will repel each other and the opposite poles will attract each other.
Mike starts by laying one magnet horizontally on the table. During each following step Mike adds one more magnet horizontally to the right end of the row. Depending on how Mike puts the magnet on the table, it is either attracted to the previous one (forming a group of multiple magnets linked together) or repelled by it (then Mike lays this magnet at some distance to the right from the previous one). We assume that a sole magnet not linked to others forms a group of its own.
<image>
Mike arranged multiple magnets in a row. Determine the number of groups that the magnets formed.
**Input**
The first line of the input contains an integer n ($1 \leq n \leq 100000$) — the number of magnets. Then n lines follow. The i-th line ($1 \leq i \leq n$) contains either characters "01", if Mike put the i-th magnet in the "plus-minus" position, or characters "10", if Mike put the magnet in the "minus-plus" position.
**Output**
On the single line of the output print the number of groups of magnets.
**Examples**
**Input**
6
10
10
10
01
10
10
**Output**
3
**Input**
4
01
01
10
10
**Output**
2

**Note**
The first testcase corresponds to the figure. The testcase has three groups consisting of three, one and two magnets.
The second testcase has two groups, each consisting of two magnets.
The program should read input from stdin and print output to stdout.
Include necessary headers and write efficient code.
Please provide only the C code in  , without any explanations or markdown formatting.
<|im_end|>
<|im_start|>assistant

Figure 16: The example prompt of CodeContests+ (Wang et al., 2025b) for evaluation test cases.

## Qwen2.5-Coder-7B-Instruct  Ours

```c
#include<stdio.h>
int main() {
    int n;
    scanf("%d", &n);
    int totalSoldiers = n;
    int killed = totalSoldiers /2;
    killed += totalSoldiers /2;
    killed += totalSoldiers /2;
    printf("%d", killed);
    return 0;
}
```

```c
#include<stdio.h>
int main(void) {
    long long n;
    (void)scanf("%lld", &n);
    long long max_killed = 0;
    max_killed = 2* n;
    (void)printf("%lld", max_killed);
    return 0;
}
```

## Unit Test

| Input | Output |
|---|---|
| 2 | 3 |
| 100000000 | 150000000 |

● **[misra-c2012-8.2]**
Function types shall be in prototype form with named parameters.

● **[misra-c2012-17.7]**
The value returned by a function having non-void return type shall be used.

Figure 17: Case Examples: Increased MISRA C Compliance with Compromised pass@1 Performance.

