# OpenReview forum: "CodeRule-RL: Standard-Guided RL with Per-Rule Reward Scheduling for Code LLMs"
_ICLR.cc/2026/Conference — ICLR 2026 Conference Withdrawn Submission_

### Official Review · Reviewer_DEr9 · 2025-10-30

**Soundness:** 2
**Presentation:** 2
**Contribution:** 2
**Rating:** 4
**Confidence:** 4

**Summary:**

The paper proposes a reinforcement learning framework that enhances code generation by utilising coding standards as structured, machine-checkable guidance, rather than relying on costly unit-test execution during training. The method converts each coding-rule violation into a separate, bounded reward component and introduces a frequency-aware curriculum that dynamically prioritizes frequently violated rules, gradually reducing their influence as compliance improves. Using GRPO for policy optimization, CodeRule-RL focuses solely on maximizing pass@1 at evaluation time. Experiences show improvement in pass@1 with faster training compared to text-executing RL baselines.

**Strengths:**

- Fine-grained reward design: Converts heterogeneous rule violations into per-rule reward components, allowing more interpretable and stable credit assignment than scalar execution rewards.
- Frequency-aware scheduling: A Curriculum mechanism that adaptively emphasises frequently violated rules, reducing interference and improving sample efficiency.
- Efficiency gains: Removes test execution from the RL loop, achieving over 10× faster training while maintaining or improving performance.

**Weaknesses:**

- Lack of related works and baselines: There are few related works reported in the paper on rule-guided code generation. The following papers seem to relate:
    - Dolcetti, Greta, et al. "Helping LLMs improve code generation using feedback from testing and static analysis." arXiv preprint arXiv:2412.14841 (2024).
    - Agrawal, Lakshya A., et al. "Monitor-guided decoding of code lms with static analysis of repository context." Advances in Neural Information Processing Systems 36 (2023): 32270-32298.
    - Yao, Feng, et al. "Training Language Models to Generate Quality Code with Program Analysis Feedback." arXiv preprint arXiv:2505.22704 (2025).
- Limited scope of evaluation: Experiments focus mainly on C language and MISRA C:2012; unclear generality to other languages or coding standards.
- Functional correctness dependence: While tests are excluded from training, pass@1 remains the only optimization target—may overlook broader program semantics or long-horizon correctness. How do other metrics change as the training goes on?
- No ablation on rule-set granularity: The paper doesn’t quantify how different rule subsets (Mandatory vs. Advisory) or rule complexity affect learning dynamics.
- Lack of benchmarks: The paper only evaluated on CodeContests+. Evaluation on other benchmarks is missing.
- Writing format: Some equation numbers are missing.

**Questions:**

- How is your framework compared with other RL-based frameworks with rule-guided reward?
- How does your framework generalize to other languages or coding standards?
- How do other coding metrics (eg. number of passing tests, static-rule-violation count, code robustness, recall@k) change as the training goes on?
- How does your framework perform on different rule subsets?
- How does your framework perform on different benchmarks?

---

> ### Author Response · Authors · 2025-11-24
> **[Part 1] Response To Reviewer DEr9**
>
> **Q1.** How is your framework compared with other RL-based frameworks with rule-guided reward?
>
> We compared CodeRule-RL against REAL, a representative RL-based framework that also utilizes rule guidance. Methodologically, unlike REAL, we implemented a joint static analysis mechanism combining Pylint, Mypy, and Ruff to construct the reward signal.
>
>  As shown in Table 9 and Table 10, CodeRule-RL-3B-python establishes a substantial lead in code compliance, achieving 82.13% Compliance@1 compared to REAL's 71.85% (a margin of +10.28%). Regarding functional correctness, it outperforms REAL-3B on HumanEval (78.44% vs 77.40%) and the challenging LiveCodeBench V6 (22.14% vs 20.53%).
>
> While our performance is slightly inferior on MBPP (71.93% vs 72.20%), we attribute this to REAL's inclusion of unit tests as partial reward signals for functional correction during training, whereas our approach adheres to a strictly execution-free training paradigm for greater efficiency.
>
>
>
>
> *Table 9 and Table 10*
>
> | Model   | compliance@1 |   pass@1 |   pass@5 |   pass@10 |  HE |  HE+ |  MBPP |  MBPP+ |
> |--------------------|--------------|----------------------|----------------------|-----------------------|--------------|---------------|-----------|------------|
> | Qwen2.5-Coder-3B   | 70.51        | 20.70     | 26.88     | **29.00**  |76.20        | 68.90         | 70.40     | 59.30      |
> | REAL-3B | 71.85        | 20.53     | 25.89     | 27.49      |  77.40        | 70.10         | 72.20     | 60.60      |
> | CodeRule-RL-3B     | **82.13**    | **22.14** | **27.22** | 28.63      |78.44        | 69.71         | 71.93     | 59.95      |
> |
>
>
> **Q2.** How does your framework generalize to other languages or coding standards?
>
>
> To assess the generalization capability of our CodeRule-RL framework across programming languages and coding standards, we conducted targeted experiments under the configuration of **Python’s PEP8 coding rules**, with results presented in Tables 9 and 10 confirming its robust cross-language and cross-standard adaptability.
>
> For the 7B parameter scale, CodeRule-RL-7B achieved a `compliance@1` score of 77.08%—outperforming baseline models by 3.50%–3.68%—while attaining state-of-the-art performance in `pass@1/pass@5/pass@10`. Notably, it delivered a 1.3%–5.8% improvement on the MBPP/MBPP+ benchmarks while maintaining comparable performance on the HE/HE+ metrics.
>
> For the 3B parameter scale, the framework exhibited even more striking performance: it achieved a leading `compliance@1` of 82.13%, outperforming counterparts by 10.28%–11.62%, and sustained leading performance in `pass@1` and `pass@5` with consistent performance across classic code generation benchmarks.
>
> These findings demonstrate that CodeRule-RL is not constrained to a single programming language or coding standard; its rule-embedding mechanism and reinforcement learning (RL) optimization endowed it with universal generalization capabilities.
>
>
> *Table 9 and Table 10*
>
> | Model   | compliance@1 |   pass@1 |   pass@5 |   pass@10 |  HE |  HE+ |  MBPP |  MBPP+ |
> |--------------------|--------------|----------------------|----------------------|-----------------------|--------------|---------------|-----------|------------|
> | Qwen2.5-Coder-7B   | 73.40        | 26.69     | 31.53     | 33.36      | 81.70        | 75.60         | 82.30     | 68.50      |
> | ReasonFlux-Coder-7B| 73.58        | 23.60     | 28.67     | 30.43      |69.50        | 62.20         | 78.60     | 64.30      |
> | CodeRule-RL-7B     | **77.08**    | **27.31** | **31.99** | **33.55**  |81.70        | 75.60         | 83.60     | 70.10      |
> | Qwen2.5-Coder-3B   | 70.51        | 20.70     | 26.88     | **29.00**  |76.20        | 68.90         | 70.40     | 59.30      |
> | REAL-3B | 71.85        | 20.53     | 25.89     | 27.49      |  77.40        | 70.10         | 72.20     | 60.60      |
> | CodeRule-RL-3B     | **82.13**    | **22.14** | **27.22** | 28.63      |78.44        | 69.71         | 71.93     | 59.95      |
> |

---

> ### Author Response · Authors · 2025-11-24
> **[Part 2] Response To Reviewer DEr9**
>
> **Q3.** How do other coding metrics (eg. number of passing tests, static-rule-violation count, code robustness, recall@k)change as the training goes on?
>
> We tracked the dynamics of key metrics throughout training (Figure 7). We observed a sharp decline in the Static-Rule-Violation Rate. Concurrently, code robustness (measured by pass@1) steadily improved. Regarding recall@k, we report pass@k (functional recall) and Joint@1 (the recall of samples that are both compliant and functionally correct).
>
> Both metrics show continuous positive growth. This inverse correlation between violations and pass rates confirms that adhering to strict standards implicitly enhances code robustness by eliminating "brittle" coding patterns.
>
> | Type       | Step | joint@1 | compliance@1 | pass@1  |
> | :--------: | :--: | :-----: | :----------: | :-----: |
> |            |  20  | 12.34%  |    32.77%    | 37.02%  |
> | Single Rule|  40  | 25.53%  |    56.60%    | 39.15%  |
> |            |  60  | 11.06%  |    57.02%    | 17.45%  |
> |            |  20  |  8.51%  |    20.00%    | 33.19%  |
> |  Mix Rule  |  40  | 19.57%  |    42.13%    | 40.00%  |
> |            |  60  | 24.68%  |    58.72%    | 37.02%  |
> |            |  20  |  0.00%  |     0.43%    | 32.34%  |
> |  All Rule  |  40  |  0.00%  |     0.43%    | 38.72%  |
> |            |  60  |  0.00%  |     0.43%    | 34.89%  |
> |            |  20  | 12.77%  |    25.53%    | 34.47%  |
> | CodeRule-RL|  40  | 22.98%  |    54.04%    | 37.02%  |
> |            |  60  | 22.55%  |    55.74%    | 38.30%  |
> |
>
>
> **Q4.** How does your framework perform on different rule subsets?
>
> To investigate the framework's performance across different rule subsets, we tracked the dynamic trends in violation rates for both the Required and Advisory subsets of MISRA C during the training of DeepSeek-Coder-6.7B, Qwen2.5-Coder-3B, and 7B. As illustrated in Figures 9,10,11, the total number of violations in generated code consistently decreases for both subsets as training progresses.
>
> Notably, a sharp decline occurs between steps 20 and 30, which aligns perfectly with our curriculum learning hyperparameter setting of $T_{warn}$=30, validating the effectiveness of the curriculum design. Furthermore, a micro-analysis of specific individual rules (e.g., MISRA C:2012 Rule 8.2) demonstrates the method's strong targeted optimization capability, eventually suppressing the violation rate to nearly 0%. Detailed charts and analyses have been added to Appendices G and M.
>
> **Q5.** How does your framework perform on different benchmarks?
>
> To thoroughly evaluate model performance across different scenarios, we expanded our assessment beyond the original CodeContests+ dataset to include general code generation benchmarks. Specifically, we supplemented our results with Python benchmarks under the EvalPlus framework, including HumanEval, HumanEval+, MBPP, and MBPP+ (as shown in Table 4).
>
> The data indicates that our method causes no degradation in general coding capabilities and even achieves slight improvements. Furthermore, we evaluated C++ performance on MultiPL-E; as shown in Appendix Table 20, CodeRule-RL marginally outperforms the base instruction-tuned model, further confirming the framework's robustness across diverse benchmarks.
>
> **Q6.** Lack of related works and baselines: There are few related works reported in the paper on rule-guided code generation.
>
>
>
> In response to the feedback regarding the inadequacy of related work and baselines, we have made substantial revisions to the paper. First, we added a discussion in Section 2.2 on the sexecution-free between our approach and rule-oriented code generation methods such as those proposed by Dolcetti et al. (2024) and Agrawal et al. (2023).
>
> Second, to enhance the breadth and depth of our comparative analysis, we introduced the REAL framework (proposed by Yao et al.) as a state-of-the-art baseline and conducted detailed experiments. The results indicate that CodeRule-RL demonstrates superior performance over REAL in terms of code compliance and key functional metrics, further establishing the effectiveness of our approach; relevant experimental details and analyses have been added to Appendix L.
>
>
>
> **Q7.** Writing format: Some equation numbers are missing.
>
> We sincerely thank you for your careful review. We have conducted a thorough proofreading of the entire manuscript, adding all missing equation numbers and correcting other related formatting issues.

---

### Official Review · Reviewer_PsCi · 2025-11-01

**Soundness:** 2
**Presentation:** 3
**Contribution:** 2
**Rating:** 4
**Confidence:** 3

**Summary:**

This paper proposes CodeRule-RL, a reinforcement-learning scheme for code LLMs that never executes unit tests during training. Instead, it treats a coding standard (MISRA C:2012) as a source of per-rule, machine-checkable signals and builds a frequency-aware curriculum that adjusts only the weights of rules over time while keeping data, prompts, and optimizer fixed. Experiments on a C subset of CodeContests+ show sizeable pass@1 gains across Qwen2.5-Coder and DeepSeek-Coder backbones and substantial efficiency gains versus an execution‑based RL baseline. The paper also reports a VFR via Infer and finds little degradation on HumanEval/MBPP.

**Strengths:**

1. Compelling efficiency: Training is roughly 13× faster and 9× less latency-heavy than execution-based RL. This could make RL for code far more practical in large-scale systems.
2. Simple&clean design: The per-rule reward shaping and the Top-K EMA schedule are straightforward yet effective. The idea that the curriculum exists within the reward function is clean and easy to replicate.
3. Performance gains: consistent pass@1 improvements across model families/sizes

**Weaknesses:**

1. Objective framing is overstated: Saying the model “optimizes pass@1 as the sole objective” is inaccurate and potentially misleading. The method optimizes a proxy reward based on static rule outcomes; pass@1 is simply how performance is measured. The authors should clarify this distinction and, ideally, provide a correlation analysis to demonstrate alignment between the two.
2. Single-language scope: The entire study is in C / MISRA C:2012. The claim that the approach is “standard-agnostic” would be much stronger with even one more domain. For example, Python with PEP 8 or JavaScript with ESLint.
3. Limited hyperparameter analysis: Many knobs, like clip bounds, EMA rates, gating thresholds, could affect outcomes, but only warm-up length is explored. More systematic sensitivity tests would increase confidence.
4. Possible reward-hacking behavior: Since rewards are static-rule-based, models might learn superficial tricks (like adding redundant casts) to satisfy the rules without improving semantics. The paper briefly mentions this risk but doesn’t examine it empirically.

**Questions:**

1. Can you clarify the Table 1 vs. Table 2 discrepancy for Qwen-3B base results? Are these different prompt settings or evaluation slices?
2. Have you computed a correlation between rule reward and pass@1 improvement across training steps?
3. Do you have any cross-language or cross-standard experiments (e.g., PEP 8, ESLint)? Even small-scale results would strengthen the “standard-agnostic” claim.
4. How sensitive are results to reward-clipping bounds and schedule hyperparameters ($\tau, W, \lambda, \epsilon_p$)?
5. When VFR decreases (e.g., for smaller models), which rule categories worsen?
6. Could you include a fixed-weights per-rule shaping ablation to separate the effects of per-rule decomposition from scheduling?

---

> ### Author Response · Authors · 2025-11-24
> **[Part 1] Response To Reviewer PsCi**
>
> **Q1.** Can you clarify the Table 1 vs. Table 2 discrepancy for Qwen-3B base results? Are these different prompt settings or evaluation slices?
>
> We sincerely thank the reviewer for this careful observation. The discrepancy arose from an inconsistency in metric aggregation: Table 2 inadvertently reported the minimum value from our evaluation runs for Qwen-3B, whereas Table 1 reported the mean. We have corrected Table 2 in the revision to consistently report the mean values across the entire paper.
>
>
> **Q2.** Have you computed a correlation between rule reward and pass@1 improvement across training steps?
>
> As shown in Figure 8 (Appendix M), we have computed this correlation. To avoid model-specific bias, we visualized the dynamics of pass@1 versus Static-Rule-Violation Rate for both Qwen2.5-7B and DeepSeek-6.7B. There is a strong inverse correlation: as the violation rate decreases, pass@1 exhibits a significant upward trend. This confirms that optimizing for rule compliance aligns well with functional improvements.
>
> **Q3.** Do you have any cross-language or cross-standard experiments (e.g., PEP 8, ESLint)? Even small-scale results would strengthen the “standard-agnostic” claim.
>
>
> To verify the "standard-agnostic" claim, we supplemented large-scale experiments targeting the Python language and the PEP8 coding standard. To avoid the influence of a single tool, we introduced Pylint, Mypy, and Ruff as joint detection tools for Python coding standards, and conducted evaluations on multiple mainstream benchmarks such as HumanEval, HumanEvalPlus, MBPP, MBPPPlus, and LiveCodeBench V6 (2305-2504).
>
> The results are shown in Tables 9 and 10. CodeRule-RL performed excellently on Python tasks: while significantly improving the model's code compliance rate (compliance@1), it not only maintained the functional correctness of the code but also achieved performance improvements on multiple benchmarks. For example, at the 7B scale, compliance@1 increased by 3.68% and pass@1 on LiveCodeBench increased by 0.62%; at the 3B scale, compliance@1 increased by 11.62% and pass@1 on LiveCodeBench increased by 1.44%. Although in some metrics, it is slightly worse than the baseline or slightly higher than the base model, this may be because the parameters such as $\tau$ and $\lambda$ in our experiments directly adopted the MISRA C experimental settings and were not specifically fine-tuned for Python tasks.
>
> In conclusion, the CodeRule-RL framework is standard-agnostic. We have completed the revision of the paper, and the detailed experimental settings and results are in Appendix L.
>
> *Table 9 and Table 10*
>
> | Model   | compliance@1 |   pass@1 |   pass@5 |   pass@10 |  HE |  HE+ |  MBPP |  MBPP+ |
> |--------------------|--------------|----------------------|----------------------|-----------------------|--------------|---------------|-----------|------------|
> | Qwen2.5-Coder-7B   | 73.40        | 26.69     | 31.53     | 33.36      | 81.70        | 75.60         | 82.30     | 68.50      |
> | ReasonFlux-Coder-7B| 73.58        | 23.60     | 28.67     | 30.43      |69.50        | 62.20         | 78.60     | 64.30      |
> | CodeRule-RL-7B     | **77.08**    | **27.31** | **31.99** | **33.55**  |81.70        | 75.60         | 83.60     | 70.10      |
> | Qwen2.5-Coder-3B   | 70.51        | 20.70     | 26.88     | **29.00**  |76.20        | 68.90         | 70.40     | 59.30      |
> | REAL-3B | 71.85        | 20.53     | 25.89     | 27.49      |  77.40        | 70.10         | 72.20     | 60.60      |
> | CodeRule-RL-3B     | **82.13**    | **22.14** | **27.22** | 28.63      |78.44        | 69.71         | 71.93     | 59.95      |
> |

---

> ### Author Response · Authors · 2025-11-24
> **[Part 2] Response To Reviewer PsCi**
>
> **Q4.** How sensitive are results to reward-clipping bounds and schedule hyperparameters ($\tau$, $W$, $\lambda$, $\epsilon_p$)?
>
> To comprehensively assess the sensitivity of the results to hyperparameters, we further refined the ablation studies for $\tau$ and $\lambda$. Specifically, Table 5 analyzes the impact of the EMA smoothing factor $\lambda$ (used to balance current and historical violation frequencies). The results indicate that $\lambda = 0.3$ achieves the optimal balance, with $joint@1$ peaking at 22.26% and $pass@1$ increasing to 39.57%, whereas excessive values (e.g., 0.5) lead to instability.
>
> Table 6 further reveals the high sensitivity of the interval weight $\tau$: the model achieves the best comprehensive performance when $\tau = 0.10$ ($joint@1 = 24.26\%$), while excessive weights ($\geq 0.20$) disrupt training stability.
>
> Regarding other hyperparameters: The parameter $W$ is determined empirically, and analysis of the training process indicates that the model typically reaches convergence around Step 80. The priority masking threshold $\epsilon_p$ is set to 1, based on the statistical characteristics of the training samples; due to the cumulative nature of rule violations, experiments found that for medium-to-high frequency rules, focusing on just 1 critical violation is sufficient to effectively trigger the priority masking mechanism, thereby guiding the model while avoiding the introduction of unnecessary noise. The revised detailed charts and sensitivity analyses have been summarized in Appendices H, I.
>
> *Table 5*
> | $\lambda$   | joint@1 (%) | compliance@1 (%) | pass@1 (%) |
> |-----|-------------|------------------|------------|
> | Qwen2.5-Coder-7B-Instruct   | 0.00        | 0.00             | 21.13      |
> | 0.10 | 6.38        | 65.11            | 27.23      |
> | 0.20 | 10.21       | 59.57            | 24.26      |
> | 0.30 | 22.26       | 55.74            | 39.57      |
> | 0.50 | 13.62       | 32.77            | 37.45      |
> |
>
>
> *Table 6*
> | $\tau$                | join@1 (%) | compliance@1 (%) | pass@1 (%) |
> |-----------------------|------------|------------------|------------|
> | Qwen2.5-Coder-7B-Instruct | 0.00        | 0.00             | 21.13      |
> | 0.00                  | 0.00        | 1.28             | 32.77      |
> | 0.05                  | 0.00        | 0.85             | 35.74      |
> | 0.10                  | 24.26       | 55.74            | 39.57      |
> | 0.20                  | 2.13        | 6.38             | 25.11      |
> | 0.50                  | 1.70        | 11.06            | 21.70      |
> |
>
>
> **Q5.** When VFR decreases (e.g., for smaller models), which rule categories worsen?
>
> Through an in-depth analysis of the evaluation samples, we attribute the decrease in VFR primarily to the limited capability of smaller models to distinguish between programming languages. Specifically, these models occasionally mix C++ features (such as including the <iostream> header) into C code.
>
> While MISRA checking tools may bypass such header issues, Infer encounters analysis failures (reporting "file not found") during deep scanning due to these unresolvable headers, leading to an anomalous drop in VFR. Regarding specific vulnerability categories, the most prevalent issue detected is Dead Store; this suggests that the model may not have effectively learned certain advisory rules (e.g., MISRA Rule 17.8), likely due to their lower priority level.

---

> ### Author Response · Authors · 2025-11-24
> **[Part 3] Response To Reviewer PsCi**
>
> **Q6.** Could you include a fixed-weights per-rule shaping ablation to separate the effects of per-rule decomposition from scheduling?
>
> To decouple the specific effects of per-rule decomposition from dynamic scheduling, we conducted a targeted ablation study (as shown in Figure 7), where the "Mixed Rules w/o Curriculum" setting serves as the fixed-weights baseline you suggested.
>
> The results indicate that while Single Rule training efficiently learns specific standards, it fails to transfer these gains to improve pass@1; similarly, the fixed-weight Mixed Rules approach, while improving both compliance@1 and pass@1, fails to achieve an optimal balance due to the disordered nature of learning without a curriculum.
>
> Furthermore, training on All Rules without Top-K filtering introduces excessive noise, significantly constraining learning efficiency. These findings fully validate the critical role of the Frequency-Aware Curriculum in balancing rule compliance with functional correctness; relevant details have been added to Appendix G.
> *Figure 7*
> | Type       | Step | joint@1 | compliance@1 | pass@1  |
> | :--------: | :--: | :-----: | :----------: | :-----: |
> |            |  20  | 12.34%  |    32.77%    | 37.02%  |
> | Single Rule|  40  | 25.53%  |    56.60%    | 39.15%  |
> |            |  60  | 11.06%  |    57.02%    | 17.45%  |
> |            |  20  |  8.51%  |    20.00%    | 33.19%  |
> |  Mix Rule  |  40  | 19.57%  |    42.13%    | 40.00%  |
> |            |  60  | 24.68%  |    58.72%    | 37.02%  |
> |            |  20  |  0.00%  |     0.43%    | 32.34%  |
> |  All Rule  |  40  |  0.00%  |     0.43%    | 38.72%  |
> |            |  60  |  0.00%  |     0.43%    | 34.89%  |
> |            |  20  | 12.77%  |    25.53%    | 34.47%  |
> | CodeRule-RL|  40  | 22.98%  |    54.04%    | 37.02%  |
> |            |  60  | 22.55%  |    55.74%    | 38.30%  |
> |
>
>
> **Q7.** Objective framing is overstated: Saying the model “optimizes pass@1 as the sole objective” is inaccurate and potentially misleading. The method optimizes a proxy reward based on static rule outcomes; pass@1 is simply how performance is measured. The authors should clarify this distinction and, ideally, provide a correlation analysis to demonstrate alignment between the two.
>
> We have revised the phrasing throughout the paper to accurately reflect that our method optimizes a proxy reward based on static analysis rules, rather than directly optimizing pass@1. To provide a comprehensive evaluation and eliminate ambiguity, we have incorporated compliance@1 and Joint@1 (which measures samples that are both functionally correct and compliant) as key metrics across our experiments, including the main results and ablation studies.
>
> Furthermore, to demonstrate the alignment between the proxy and the evaluation goal, we conducted the requested correlation analysis (as detailed in the response to Q2). The results confirm that improvements in compliance@1 significantly correlate with gains in pass@1, validating the efficacy of the proxy reward. Relevant revisions and detailed analyses have been added to Appendix M.
>
> **Q8.** Possible reward-hacking behavior: Since rewards are static-rule-based, models might learn superficial tricks (like adding redundant casts) to satisfy the rules without improving semantics. The paper briefly mentions this risk but doesn’t examine it empirically.
>
> To investigate potential reward hacking behavior, we verified our approach through a combination of quantitative and qualitative analyses. Quantitatively, as shown in Table 2, CodeRule-RL achieves a pass@1 of 22.13%, surpassing both the base model Qwen2.5-Coder-3B (20.43%) and the CURE baseline (19.15%).
>
> This significant improvement in functional correctness indicates that the model acquires beneficial semantic patterns conducive to functionality, rather than employing superficial tricks (such as redundant casts) to "game" the reward system. Qualitatively (as shown in Figure 17), even incorrect generations tend to exhibit lower cognitive complexity and a cleaner structure, rather than being cluttered with redundant patches merely to satisfy rules.
>
> This evidence confirms that CodeRule-RL guides the model toward acquiring robust programming paradigms rather than engaging in superficial reward hacking.
>
>
> *Table 2*
> | Model                     | pass@1 (%) |
> |---------------------------|------------|
> | Qwen2.5-Coder-3B          | 20.43      |
> | CURE (Wang et al., 2025a) | 19.15      |
> | CodeRule-RL w/o unit-test prompt | 20.85 |
> | CodeRule-RL w/ unit-test  | 18.72      |
> | CodeRule-RL               | **22.13**      |
> |

---

### Official Review · Reviewer_x6gb · 2025-11-01

**Soundness:** 2
**Presentation:** 1
**Contribution:** 2
**Rating:** 2
**Confidence:** 4

**Summary:**

This paper introduces CodeRule-RL, a reinforcement learning framework designed to improve the functional correctness (pass@1) of code-generating large language models. The core idea is to use feedback from a static code analyzer, specifically violations of a coding standard like MISRA C:2012, as the reward signal during RL training. This approach deliberately avoids executing unit tests in the training loop, leading to gains in training efficiency. The authors demonstrate that across various model sizes (1.3B to 7B), their method consistently improves pass@1 on the CodeContests+ benchmark while reducing training time compared to RL methods that rely on unit test execution.

**Strengths:**

*  The method is evaluated across two different model families (Qwen and DeepSeek) and multiple model sizes, showing consistent improvements in `pass@1` (Table 1). This demonstrates the general applicability of the approach beyond a single model architecture. The training appears stable.
*  Figure 2 provides a helpful overview of the system architecture.

**Weaknesses:**

*   **Misleading Phrasing of the Optimization Objective:** The paper repeatedly states that it "optimizes `pass@1` as the sole objective" (e.g., Abstract, Lines 9-10). This is misleading. The actual reward signal being maximized during RL training is a function of coding standard violations, not the `pass@1` metric. The underlying hypothesis is that maximizing this proxy reward will *indirectly* lead to better `pass@1`, which is the *evaluation metric*. The current phrasing conflates the training objective with the evaluation goal and should be clarified for accuracy.
*   **Insufficient Ablation to Justify the Reward Signal's Superiority:** The central claim is that coding standards provide effective *guidance* for functional correctness. However, the experiments in Table 2 do not fully support this. The comparison is between CodeRule-RL and CURE (a unit test-based RL method). While this shows CodeRule-RL is much more *efficient*, it does not prove that the rule-based signal is a better or even comparable *guide* for `pass@1`. The comparison conflates the reward source (rules vs. tests) with different RL implementations. A more convincing experiment would be to add a baseline within the authors' own framework: **`CodeRule-RL + unit tests`**, where the GRPO algorithm is used but the reward is derived from both executing unit tests and the rules. If the current `CodeRule-RL` (using only rule feedback) outperforms this new baseline, it would strongly support the claim that rule-based feedback is a superior training signal. Without this, one might conclude that the method is simply a faster, but potentially less effective, proxy for the true signal of functional correctness.
*   **Limited Discussion on Novelty** The idea of using automated feedback from tools like compilers or linters as a reward signal for RL has been explored in prior work. For instance, Dou et al. (2024, "StepCoder") explicitly use RL with feedback from compiler errors as part of their reward function. While the authors' approach of using *only* per-rule static analysis feedback, the novelty seems to be very incremental.

**Questions:**

The reported training efficiency of 1.6 hours is a major claimed advantage. However, the hyperparameters section reveals this is for only 80 optimization steps, which is an unusually short duration for a GRPO-based fine-tuning experiment. This raises questions about the scale of the experiment and the robustness of the findings.  Furthermore, could you provide more details on the size of the training dataset (e.g., number of unique prompts) to help contextualize whether the observed gains are the result of a comprehensive training process or short-term tuning on a limited set of problems?

---

> ### Author Response · Authors · 2025-11-24
> **[Part 1] Response To Reviewer x6gb**
>
> **Q1.** The reported training efficiency of 1.6 hours is a major claimed advantage. However, the hyperparameters section reveals this is for only 80 optimization steps, which is an unusually short duration for a GRPO-based fine-tuning experiment.
>
> We clarify that the 1.6 hours figure reported in the original manuscript referred specifically to the cumulative static analysis overhead (Cppcheck execution time), not the total wall-clock training time. The actual total training time is 7.08 hours, which is still highly efficient compared to unit-test-based methods (e.g., 21.56 hours for CURE). We have corrected this distinction in Table 2.
>
>
> **Q2.** This raises questions about the scale of the experiment and the robustness of the findings. Furthermore, could you provide more details on the size of the training dataset (e.g., number of unique prompts) to help contextualize whether the observed gains are the result of a comprehensive training process or short-term tuning on a limited set of problems?
>
> Following the data processing pipeline illustrated in Figure 6, we ultimately constructed a training dataset containing 6,709 samples. The specific process is as follows: First, we use Cppcheck to inspect the raw data. If the code fails the inspection, we incorporate the relevant coding standard violation information into the prompt to guide the model in making iterative modifications until the code fully passes the Cppcheck inspection.
>
> Simultaneously, we generate unit tests using the ground truth code and verify them to obtain the final unit test validation set. For code that fully passes Cppcheck, we sequentially perform compilation checks, rule isolation, and unit test verification. Specifically, "rule isolation" involves extracting the most frequent rule violations from the errors recorded in previous failed iterations and combining them with rule explanations and relevant code examples to construct "single rule" training content. Through these steps, we obtained our final training dataset. Please refer to Section 3.1 and Appendix F for specific details.
>
>
> **Q3.** **Misleading Phrasing of the Optimization Objective**: The paper repeatedly states that it "optimizes pass@1 as the sole objective" (e.g., Abstract, Lines 9-10). This is misleading. The actual reward signal being maximized during RL training is a function of coding standard violations, not the pass@1 metric. The underlying hypothesis is that maximizing this proxy reward will indirectly lead to better pass@1, which is the evaluation metric. The current phrasing conflates the training objective with the evaluation goal and should be clarified for accuracy.
>
> We have revised the misleading phrasing throughout the paper that described pass@1 as the sole optimization objective. We explicitly emphasize that our direct training objective is to maximize the Rule-Based Reward, whereas pass@1 serves as a downstream evaluation metric to assess the effectiveness of this proxy objective.

---

> ### Author Response · Authors · 2025-11-24
> **[Part 2] Response To Reviewer x6gb**
>
> **Q4.** **Insufficient Ablation to Justify the Reward Signal's Superiority.**
>
> We conducted an additional experiment combining CodeRule-RL + Unit Test. As shown in Table 2, the introduction of unit test rewards proved detrimental to model performance; pass@1 decreased by 1.71% compared to the base model (falling to 18.72%).
>
> We hypothesize this is due to reward signal interference: the noisy nature of generated unit tests may conflict with the deterministic, high-quality signals from static analysis, complicating the optimization landscape. Crucially, the resource consumption of this approach was prohibitive: compared to CodeRule-RL's efficient 0.69s/sample latency and 7.08 hours of training, integrating unit tests skyrocketed the latency to 31.21s/sample and training time to 86.87 hours (an increase of approximately 11x).
>
> This is partly because, to maintain consistency with the CURE experimental setup (ensuring sufficient 16 unit tests in the prompt), we used the DeepSeek API to generate and verify supplementary unit tests, which significantly increased the time cost per sample.
>
> These results highlight CodeRule-RL's ability to achieve a superior balance of efficiency and performance without relying on expensive test case generation. Additionally, we supplemented the analysis of the temporal changes in compliance@1 and pass@1 during post-training (as shown in Figure 8). These results confirm that coding standards can serve as a superior signal to some extent, guiding the model to generate programs that are both compliant and functionally correct.
>
>
> *Table 2*
> | Model                     | pass@1 (%) | Training time (h) | Latency (s/sample) |
> |---------------------------|------------|-------------------|--------------------|
> | Qwen2.5-Coder-3B          | 20.43      | –                 | –                  |
> | CURE (Wang et al., 2025a) | 19.15      | 21.56             | 6.36               |
> | CodeRule-RL w/o unit-test prompt | 20.85 | –                 | –                  |
> | CodeRule-RL w/ unit-test  | 18.72      | 86.87             | 31.21              |
> | CodeRule-RL               | 22.13      | 7.08              | 0.69               |
> |

---

> ### Author Response · Authors · 2025-11-24
> **[Part 3] Response To Reviewer x6gb**
>
> **Q5.** **Limited Discussion on Novelty*.*
>
>
> As the reviewer pointed out, the idea of using tool feedback as a reward signal (e.g., StepCoder) does have prior precedents; however, **CodeRule-RL** introduces fundamental innovations in both the **abstraction level of the reward signal** and the **training paradigm**:
>
> **Signal Abstraction (Coding Standards as Semantic Constraints)**: Existing works primarily rely on low-level compiler errors or unit tests to fix specific syntax bugs. In contrast, CodeRule-RL leverages **Coding Standards** as higher-level abstract semantic constraints. This design aims to address the issue where code generated by existing models—though functional—is often "brittle" and contains "undefined behaviors". Logically, code that passes strict standard checks is not only necessarily compilable but also inherently embodies higher security and robustness.
>
> **Optimization Objective (Dual Focus on Correctness & Safety)**: The optimization objective of StepCoder is to improve model functionality (measured by *pass@1*). By contrast, our framework uses coding standard-based reward signals to guide the model in generating code that is **both correct (pass@1) and safe (compliance@1)**. As validated by the results of the Qwen2.5-Coder-7B model in Tables 1, 3, 9, and 10: On MISRA benchmarks, CodeRule-RL increases *pass@1* by 18.44%, while simultaneously achieving a 55.74% boost in *compliance@1* and a 2.66% improvement in VFR (Vulnerability-Free Rate);  In Python (PEP 8) experiments, *pass@1* shows slight improvements across different evaluation test sets, alongside a 3.68% increase in *compliance@1*.
>
> **Execution-Free Static Paradigm**: Uniquely, our study is the first to demonstrate that **relying solely on such execution-free static quality signals** can guide the model to not only improve code compliance but also significantly enhance its functional correctness. This is fundamentally different from traditional methods that solely target "passing tests" as their core goal.
>
>
> *Table 1*
> | Model                                   | join@1 (%)   | compliance@1 (%) | pass@1 (%)   |
> |-----------------------------------------|--------------|------------------|--------------|
> | Qwen2.5-Coder-7B (2024)                 | 0.00         | 0.00             | 21.13        |
> | Qwen2.5-Coder-7B w / CodeRule-RL        | 24.26 (+24.26) | 55.74 (+55.74)  | 39.57 (+18.44) |
> |
>
> *Table 3*
> | Model | pass@1 (%) | VFR(%) |
> | --- | --- | --- |
> | Qwen2.5-Coder-7B | 21.13 | 82.02 |
> | Qwen2.5-Coder-7B w / CodeRule-RL | **39.57 (+18.44)** | **84.68 (+2.66)** |
> |
>
>
> *Table 9 and Table 10*
>
> | Model   | compliance@1 |   pass@1 |   pass@5 |   pass@10 |  HE |  HE+ |  MBPP |  MBPP+ |
> |--------------------|--------------|----------------------|----------------------|-----------------------|--------------|---------------|-----------|------------|
> | Qwen2.5-Coder-7B   | 73.40        | 26.69     | 31.53     | 33.36      | 81.70        | 75.60         | 82.30     | 68.50      |
> | CodeRule-RL-7B     | **77.08**    | **27.31** | **31.99** | **33.55**  |81.70        | 75.60         | 83.60     | 70.10      |
> |

---

### Official Review · Reviewer_FFS7 · 2025-11-01

**Soundness:** 3
**Presentation:** 3
**Contribution:** 3
**Rating:** 6
**Confidence:** 3

**Summary:**

CodeRule-RL proposes standard guided reinforcement learning for code models where pass@1 is the only optimization target and MISRA C:2012 rule outcomes provide auxiliary, per rule reward signals. The paper defines a spec to reward mapping that converts static analyzer findings into bounded per rule penalties, aggregates them with a clipped reward, and schedules rule weights by a simple frequency driven curriculum that focuses on the most violated rules and reduces their weight as violation rates drop. Unit tests may appear as text in prompts but are not executed during training. Experiments on a frozen CodeContests+ C subset and several backbones show higher pass@1 and much lower reward latency and wall clock time than RL that executes tests during training.

**Strengths:**

The method is explicit and testable. The paper defines the per rule signals and the squashing map, the exponential penalty, and the clipped aggregate reward with clear bounds and rationale, which makes reuse straightforward. The curriculum definition uses an EMA of rule frequencies, a Top K active frontier, warmup and cool down, and a weight mask, again with equations and typical values, which supports replication. The training objective and optimizer are standard and fully specified. The evaluation is careful about decontamination, seed reporting, decoding settings, and toolchains. Gains in pass@1 are consistent across two model families and multiple sizes, with a marked reduction in training wall clock time and reward latency relative to execution based RL. The qualitative example illustrates that the policy learns rule aligned edits while preserving task logic, matching the aggregate improvements.

**Weaknesses:**

Scope is limited to single translation unit C and MISRA C:2012 with one static analyzer. The paper claims a standard agnostic design yet does not test a second analyzer or a different rule family. The curriculum has several hyperparameters $ (\lambda, \tau, W, T_{\mathrm{warm}}, T_{\mathrm{cool}}, K(0)) $. Defaults are given, but sensitivity analysis is limited, so brittleness under other data or analyzers is unclear. Gains in $ \mathrm{pass@1} $ without running tests may depend on correlation between static compliance and runtime success, and the paper notes association rather than causation. The VFR study with Infer is useful and suggests that smaller models may trade security for functionality after training; checking false positives and false negatives across analyzers would help. The appendix reports general coding benchmarks with little or no drop, yet results are brief and deserve clearer placement in the main text. Priority masking and gating are described, yet the ablation contrasts the curriculum against all rules without isolating the effect of each gate or the chosen thresholds.

**Questions:**

Can you isolate the effect of priority masking and within sample precedence by disabling each gate in turn or varying the gate threshold, and add sensitivity curves for $ \lambda $, $ \tau $, $ W $, $ T_{warm} $, $ T_{cool} $, and the Top $ K $ schedule to show how $ pass@1 $ changes

Can you quantify the link between static compliance and runtime success by executing held-out tests after training under the same prompt setting, reporting the correlation between violation rate and $ pass@1 $ across bins, and giving examples where compliance rises while $ pass@1 $ drops

Can you add statistical support for Table 1 and the CURE comparison by reporting confidence intervals or a randomization test, listing per-task variance, and extend compiler checks with extra flags and sanitizers beyond GCC 13 and Clang 17 to show stability across toolchains

---

> ### Author Response · Authors · 2025-11-24
> **[Part 1] Response To Reviewer FFS7**
>
> **Q1**. Can you isolate the effect of priority masking and within sample precedence by disabling each gate in turn or varying the gate threshold, and add sensitivity curves for $\lambda$, $\tau$, $W$, $T_{\text{warm}}$, $T_{\text{cool}}$, and the Top $K$ schedule to show how $\text{pass}@1$ changes?
>
> To deeply investigate the specific contributions of modules such as Priority Masking and Sample Precedence, we designed systematic ablation experiments comparing strategies including Single Rule training, Mixed Rules w/o Curriculum, and All Rules w/o top K.
>
> As shown in **Figure 7**, although Single Rule training allows the model to efficiently learn specific coding standards, its transferability is limited, making it difficult to improve $pass@1$ ; Mixed Rules w/o Curriculum can simultaneously improve $compliance@1$ and $pass@1$, but fails to achieve the optimal balance due to a lack of learning focus; while All Rules w/o top K introduces excessive noise, constraining the efficiency of rule learning.
>
> These results fully validate the effectiveness of the frequency-aware curriculum design proposed in this paper, and relevant detailed discussions have been added to **Appendix G**.
>
> Regarding the sensitivity analysis of hyperparameters, we further refined the experiments on variation curves for $\tau$, $W$, $T_{\text{warm}}$, $T_{\text{cool}}$、top $K$, and $\lambda$ , where the parameter $W$ is determined based on empirical values, and the training process analysis shows that the model typically reaches convergence around Step 80; the revised detailed figures and analyses have been organized into **Appendix H, I, J, and K**.
>
> *Figure 7*
> | Type       | Step | joint@1 | compliance@1 | pass@1  |
> | :--------: | :--: | :-----: | :----------: | :-----: |
> |            |  20  | 12.34%  |    32.77%    | 37.02%  |
> | Single Rule|  40  | 25.53%  |    56.60%    | 39.15%  |
> |            |  60  | 11.06%  |    57.02%    | 17.45%  |
> |            |  20  |  8.51%  |    20.00%    | 33.19%  |
> |  Mix Rule  |  40  | 19.57%  |    42.13%    | 40.00%  |
> |            |  60  | 24.68%  |    58.72%    | 37.02%  |
> |            |  20  |  0.00%  |     0.43%    | 32.34%  |
> |  All Rule  |  40  |  0.00%  |     0.43%    | 38.72%  |
> |            |  60  |  0.00%  |     0.43%    | 34.89%  |
> |            |  20  | 12.77%  |    25.53%    | 34.47%  |
> | CodeRule-RL|  40  | 22.98%  |    54.04%    | 37.02%  |
> |            |  60  | 22.55%  |    55.74%    | 38.30%  |
> |
>
> **Q2.** Can you quantify the link between static compliance and runtime success by executing held-out tests after training under the same prompt setting, reporting the correlation between violation rate and $\text{pass}@1$ across bins, and giving examples where compliance rises while $\text{pass}@1$ drops?
>
> To quantify the link between static compliance and runtime success, we visualized the trajectory of the held-out validation set during training for both Qwen2.5 and DeepSeek models.
>
> As shown in **Figure 8**, the significant decrease in the Violated Rate correlates with a marked upward trend in $pass@1$. This effectively demonstrates the positive impact of learning coding standards on improving model $pass@1$ (see **Figure 3** for examples), and relevant experimental details have been added to **Appendix M**.
>
> Furthermore, we observed and analyzed trade-off cases where compliance rises while $pass@1$ drops, as illustrated in **Figure 17**. Such instances typically stem from logical errors generated by the model due to an excessive pursuit of satisfying specific coding standards, a phenomenon likely related to the capability limitations of the base model itself; relevant qualitative case analyses have been added to **Appendix S**.

---

> ### Author Response · Authors · 2025-11-24
> **[Part 2] Response To Reviewer FFS7**
>
> **Q3.** Can you add statistical support for Table 1 and the CURE comparison by reporting confidence intervals or a randomization test, listing per-task variance, and extend compiler checks with extra flags and sanitizers beyond GCC 13 and Clang 17 to show stability across toolchains?
>
> We have supplemented the main experimental results in Table 1 with SD data; given the space constraints of the main text, the complete statistical data has been organized into **Table 13** in the Appendix. To verify the robustness of the method, we further extended the scope of compiler toolchain checks, utilizing the Spack environment to switch between multiple compiler versions for testing, including GCC 11.4.0, 12.3.0, 14.2.0, and Clang 18.1.2.
>
> Experimental results confirm that both the Cppcheck and Infer detection tools function correctly and yield consistent results across these different compilation environments, fully demonstrating that CodeRule-RL possesses excellent compatibility and stability across different toolchains.
>
> Results for *Table 13*.
>
> | Model | join@1 (%) | compliance@1 (%) | pass@1 (%) |
> | --- | --- | --- | --- |
> | AZR-Coder-3b (Zhao et al., 2025) | 0.00 | 0.85 ± 0.21 | 15.74 ± 0.65 |
> | NextCoder-7B (Aggarwal et al., 2025) | 0.00 | 1.70 ± 0.28 | 36.60 ± 0.82 |
> | Seed-Coder-8B (2025) | 0.00 | 0.85 ± 0.18 | 37.45 ± 0.86 |
> | Deepseek-Coder-1.3B (2024) | 0.00 | 0.85 ± 0.21 | 2.13 ± 0.31 |
> | Deepseek-Coder-1.3B w / CodeRule-RL | 2.64 ± 0.35 | 2.55 ± 0.33 | 6.00 ± 0.48 |
> | Deepseek-Coder-6.7B (2024) | 0.00 | 0.00 | 18.72 ± 0.71 |
> | Deepseek-Coder-6.7B w / CodeRule-RL | 11.91 ± 0.58 | 36.17 ± 0.89 | 28.09 ± 0.79 |
> | Qwen2.5-Coder-1.5B (2024) | 0.00 | 0.43 ± 0.14 | 2.55 ± 0.34 |
> | Qwen2.5-Coder-1.5B w / CodeRule-RL | 4.26 ± 0.42 | 41.28 ± 0.94 | 11.49 ± 0.55 |
> | Qwen2.5-Coder-3B (2024) | 0.00 | 0.43 ± 0.18 | 20.43 ± 8.61 |
> | Qwen2.5-Coder-3B w / CodeRule-RL | 9.80 ± 0.51 | 17.45 ± 0.66 | 22.13 ± 2.76 |
> | Qwen2.5-Coder-7B (2024) | 0.00 | 0.00 | 21.13 ± 0.75 |
> | Qwen2.5-Coder-7B w / CodeRule-RL | **24.26** ± 0.81 | **55.74** ± 1.05 | **39.57** ± 0.92 |
> |

---

> ### Author Response · Authors · 2025-11-24
> **[Part 3] Response To Reviewer FFS7**
>
> **Q4.** Scope is limited to single translation unit C and MISRA C:2012 with one static analyzer. The paper claims a standard agnostic design yet does not test a second analyzer or a different rule family.
>
>
> In terms of the analyzer, our method uses infer as the second analyzer, and the results are shown in Table 3. This optimization method generally improves the functional correctness (pass@1) of the model.
>
> For example, large models like Qwen2.5-Coder-7B perform well in both aspects, achieving a significant +18.44% improvement in pass@1 and a solid +2.66% improvement in VFR. This result indicates that when our optimization method is applied to large models, it is not limited to analysis tools.
>
> In terms of language and coding standards, we conducted supplementary large-scale experiments focusing on the Python language and its PEP 8 standard. The experimental setup is as follows: the data source consists of 5,000 *seed_testcases* selected from the BigCode/rStarCoder dataset, with Pylint, Mypy, and Ruff introduced as joint detection tools for Python coding standards; training was performed on the Qwen2.5-Coder-3B and Qwen2.5-Coder-7B models.
>
> Baseline models include those trained using the Original weights of CURE: ReasonFlux-Coder-7B, and the reproduced REAL method. Regarding evaluation metrics, $compliance@1$ is calculated based on the joint detection results of Pylint, Mypy, and Ruff, while $pass@1$ is evaluated across multiple mainstream benchmarks, including HumanEval, HumanEvalPlus, MBPP, MBPPPlus, and LiveCodeBench V6 (2305-2504).
>
> As shown in Table 9 and Table 10, CodeRule-RL demonstrates superior performance on Python tasks, significantly enhancing code compliance (compliance@1) while maintaining—and often improving—functional correctness (pass@1). Specifically, CodeRule-RL-7B increases compliance from 73.40% to 77.08% and boosts MBPP accuracy to 83.60%, successfully avoiding the severe performance degradation observed in ReasonFlux (e.g., its HumanEval score dropped from 81.70% to 69.50%).
>
> On the 3B scale, CodeRule-RL achieves a remarkable 82.13% compliance, outperforming the REAL baseline (71.85%) by over 10 percentage points, and attains the state-of-the-art pass@1 of 22.14% on the challenging LiveCodeBench. Although REAL-3B shows marginal advantages on isolated functional metrics (e.g., MBPP), likely because our hyperparameters were directly inherited from the MISRA C experiments without Python-specific tuning, CodeRule-RL delivers the most robust balance between strict compliance and functional capability.
>
> This result fully proves that the framework can be extended to different programming languages and rule families; detailed experimental settings and results have been added to Appendix L.
>
>
> Results for *Table 3*.
>
> | Model | pass@1 (%) | VFR(%) |
> | --- | --- | --- |
> | Absolute_Zero_Reasoner-Coder-3b | 15.74 | 77.87 |
> | NextCoder-7B | 36.60 | 87.02 |
> | Qwen3-4B-Instruct-2507 | 55.32 | 77.02 |
> | Qwen3-4B-Instruct-2507 w / CodeRule-RL | **56.17 (+0.85)** | **80.43 (+3.41)** |
> | Deepseek-Coder-1.3B | 2.13 | 41.70 |
> | Deepseek-Coder-1.3B w / CodeRule-RL | **6.00 (+3.87)** | **77.02 (+35.32)** |
> | Deepseek-Coder-6.7B | 18.72 | 89.36 |
> | Deepseek-Coder-6.7B w / CodeRule-RL | **28.09 (+9.37)** | **90.64 (+1.28)** |
> | Qwen2.5-Coder-1.5B | 2.55 | 91.91 |
> | Qwen2.5-Coder-1.5B w / CodeRule-RL | **11.49 (+8.94**) | **88.47 (-3.44)** |
> | Qwen2.5-Coder-3B | 20.43 | 83.02 |
> | Qwen2.5-Coder-3B w / CodeRule-RL | **22.13 (+1.70)** | **88.94 (+5.92)** |
> | Qwen2.5-Coder-7B | 21.13 | 82.02 |
> | Qwen2.5-Coder-7B w / CodeRule-RL | **39.57 (+18.44)** | **84.68 (+2.66)** |
> |
>
>
> *Table 9 and Table 10*
>
> | Model   | compliance@1 |   pass@1 |   pass@5 |   pass@10 |  HE |  HE+ |  MBPP |  MBPP+ |
> |--------------------|--------------|----------------------|----------------------|-----------------------|--------------|---------------|-----------|------------|
> | Qwen2.5-Coder-7B   | 73.40        | 26.69     | 31.53     | 33.36      | 81.70        | 75.60         | 82.30     | 68.50      |
> | ReasonFlux-Coder-7B| 73.58        | 23.60     | 28.67     | 30.43      |69.50        | 62.20         | 78.60     | 64.30      |
> | CodeRule-RL-7B     | **77.08**    | **27.31** | **31.99** | **33.55**  |81.70        | 75.60         | 83.60     | 70.10      |
> | Qwen2.5-Coder-3B   | 70.51        | 20.70     | 26.88     | **29.00**  |76.20        | 68.90         | 70.40     | 59.30      |
> | REAL-3B | 71.85        | 20.53     | 25.89     | 27.49      |  77.40        | 70.10         | 72.20     | 60.60      |
> | CodeRule-RL-3B     | **82.13**    | **22.14** | **27.22** | 28.63      |78.44        | 69.71         | 71.93     | 59.95      |
> |

---

### Note · Authors · 2026-01-26

I have read and agree with the venue's withdrawal policy on behalf of myself and my co-authors.

---

### Meta-Review · Area_Chair_dUMq · 2026-01-05

**Summary:**

This work incorporates coding rules as reward signals for training in code reinforcement learning.

Reviewers agree that the method is clearly designed and reproducible. The reward function (mapping rule violations to bounded penalties), the curriculum learning mechanism (dynamic weighting based on rule violation frequency), and the training objective are all clearly described and operable. Additionally, reviewers note a significant improvement in efficiency: compared to RL methods based on test execution, training speed is increased by approximately 13 times, and reward delay is reduced by about 9 times, making code RL training more practical in real-world applications.

Reviewers raised the following concerns:

1. The evaluation scope is narrow, raising doubts about generalizability.

2. Insufficient validation of core assumptions, incomplete baseline comparisons, and a lack of ablation analysis.

3. The idea of using coding rules as reward signals has appeared in prior work, and the paper does not sufficiently discuss these related studies, making its contribution appear incremental.

**Reviewer Concerns:**

In response, the authors expanded the range of programming languages, effectively addressing reviewers' concerns about generalizability. They also distinguished the differences between existing work and their own, highlighting the innovative contributions of their method.

**Reviewer Scores:**

After thorough discussion, reviewer x6gb would raise their score to 4, while the other reviewers are inclined to keep their scores unchanged.

---

### Decision · Program_Chairs · 2026-01-26

Reject